# AMPK acts as a molecular trigger to coordinate glutamatergic signals and adaptive behaviours during acute starvation

**Moloud Ahmadi, Richard Roy\***

Department of Biology, McGill University, Montreal, Canada

**Abstract** The stress associated with starvation is accompanied by compensatory behaviours that enhance foraging efficiency and increase the probability of encountering food. However, the molecular details of how hunger triggers changes in the activity of neural circuits to elicit these adaptive behavioural outcomes remains to be resolved. We show here that AMP-activated protein kinase (AMPK) regulates neuronal activity to elicit appropriate behavioural outcomes in response to acute starvation, and this effect is mediated by the coordinated modulation of glutamatergic inputs. AMPK targets both the AMPA-type glutamate receptor GLR-1 and the metabotropic glutamate receptor MGL-1 in one of the primary circuits that governs behavioural response to food availability in *C. elegans*. Overall, our study suggests that AMPK acts as a molecular trigger in the specific starvation-sensitive neurons to modulate glutamatergic inputs and to elicit adaptive behavioural outputs in response to acute starvation.

**\*For correspondence:** richard. roy@mcgill.ca

**Competing interests:** The authors declare that no competing interests exist.

## Introduction

Most organisms are faced with unpredictable fluctuations in their natural environment that often lead to periods of limited food resources. Their ability to adapt to these changes in resource availability is critical for survival and is often a driving force in evolution (*Gray et al., 2004*; *Wang et al., 2005*). When resources are scarce, pathways associated with energy conservation at both the organismal and the cellular levels become activated, and these are often complemented by behavioural modifications that simultaneously enhance foraging efficiency (*Wang et al., 2005*; *Ashrafi, 2006*).

The mammalian central nervous system (CNS) integrates internal and external cues that signal energy demand and availability and coordinately regulates outputs ranging from energy expenditure to feeding and associated locomotory behaviours (*Cone, 2005*; *Balthasar et al., 2005*; *Belgardt et al., 2009*; *Dietrich and Horvath, 2011*; *Yang et al., 2011*; *Aponte et al., 2011*; *Sternson et al., 2013*; *Dietrich et al., 2015*). In mice, the Agouti-related protein (AGRP)- and Pro-opiomelanocortin (POMC)-expressing neurons in the arcuate nucleus of the hypothalamus form a core circuit to regulate food intake and energy expenditure through the modulation of their neuronal activity in response to hormonal signals linked to metabolic status (*Cowley et al., 2001*; *Bewick et al., 2005*; *Yang et al., 2011*). The activity of both of these neuronal populations is mediated by engaging signalling pathways that control the strength and/or plasticity of rapid, excitatory glutamatergic transmission (*Bito et al., 2010*; *Collingridge et al., 2010*; *Liu et al., 2012*), but how energy stress results in changes in neuronal activity to elicit adaptive, or even compulsive behaviours are just now beginning to be elucidated (*Dietrich et al., 2015*).

The many signalling networks that are triggered throughout the nervous system that mediate the action of small molecules, hormones and nutrients on energy balance are of major interest due to

**eLife digest** Animals often need to adapt to changes in food availability in order to survive. When food is in short supply and animals are starving, their energy reserves are low. To conserve energy, behaviours that are not essential to survival, like mating, are put on hold. Instead, animals channel their energies into foraging strategies that may help them find new food sources. These behavioural changes are likely to be caused by changes in brain activity triggered by starvation.

It is not entirely clear how starvation changes the brain and consequently how an animal behaves. It is also difficult to study how the brain regulates behaviour in response to environmental changes like food availability in larger animals with more complex nervous systems. Instead, scientists often study less complex animals like a type of worm called *C. elegans*, because this model organism has a simpler nervous system and it is easier to observe its feeding behaviours. Previous observations have revealed that well-fed worms travel backwards when they are hungry, revisiting sites where they have previously found food. Yet, when the worms are starving, they move forward more frequently, presumably to find new sources of food.

Now, Ahmadi and Roy show starving worms activate an enzyme called AMP-activated protein kinase (or AMPK for short). Worms genetically engineered to lack this enzyme tend to move backward when they are starved, instead of moving forward like typical starving worms. This shows that AMPK triggers a wider search for new food sources. Further experiments showed that AMPK acts to inhibit two receptors, which in turn, affects the activity of two different neurons. These two neurons work together to change the animal's behaviour and boost the likelihood the animal will be able to find a new food source when food is scarce.

More complex animals, including humans, also have the receptors and brain cells targeted by AMPK in response to starvation. Future studies are needed to determine whether a similar chain of events occurs in creatures with more complicated nervous systems.

their implication in, or treatment of various disorders. One of the key factors of paramount importance for metabolic homeostasis and survival is a highly conserved heterotrimeric protein kinase called AMP-activated protein kinase (AMPK). AMPK is regulated by the ratio of cellular AMP/ATP and by upstream activating kinases (*Hardie, 2008*; *Hardie et al., 2012*). It functions as a 'fuel gauge' to monitor cellular energy status by inhibiting anabolic pathways and activating catabolic pathways so as to generate sufficient levels of metabolic substrates required to maintain a minimal threshold of basal cellular activities (*Hardie, 2008*; *Hardie et al., 2012*; *Hardie and Ashford, 2014*)

Energy stress has been demonstrated to induce adaptive behaviours in a neuronal AMPK-dependent manner (*Lee et al., 2008*; *Cunningham et al., 2014*). Moreover, accumulating evidence has implicated AMPK in the hypothalamic regulation of metabolic rate and food intake behaviour (*Kola, 2008*; *Lopez et al., 2010*; *Lim et al., 2010*; *Yang et al., 2011*; *Schneeberger and Claret, 2012*). However, our understanding of how starvation influences adaptive foraging behaviours in an AMPK-dependent manner is still largely unknown, mostly due to the overwhelming complexity of the response in higher animals (*Dietrich et al., 2015*).

In *C. elegans* these foraging behaviours are comparatively simple, consisting of a series of forward or backward movements, specific turns, or changes in direction (*Gray et al., 2005*; *Piggott et al., 2011*; *Chen et al., 2013*; *Hendricks, 2015*). Food availability has been demonstrated to affect various aspects of these key elements in *C. elegans* locomotion (*Sawin et al., 2000*; *Gray et al., 2005*; *Chalasani et al., 2007*; *Flavell et al., 2013*). In the absence of food, well-fed animals reverse frequently, a behavioural pattern that reflects a sensory memory of food that is expressed by the navigation circuit and results in efficient exploration of a limited area. In contrast, starvation suppresses reversals and induces forward movement (runs) that allow animals to explore distal areas; a strategy that is referred to as dispersal behaviour or alternatively, distal exploration (*Gray et al., 2005*).

The simplicity of the neural circuits and locomotory behaviours in conjunction with its amenability to genetic manipulation makes *C. elegans* an ideal model to investigate the mechanisms through which AMPK regulates neuronal activity and adaptive locomotory behaviour (searching for food) in

response to hunger. Unlike most other organisms studied to date, *C. elegans* mutants that lack all AMPK signalling are viable, but show clear phenotypes when subjected to energy stress (*Narbonne and Roy, 2006*; *2009*). Therefore, using calcium imaging, cell type–specific optogenetic techniques, and classic genetic analysis, we identified and characterized the neural circuit in which AMPK functions as a molecular switch. This circuit includes the AIB and AIY interneurons; two neurons that form one of the primary circuits that dictate appropriate food- and odour-evoked behaviours (*Gray et al., 2005*; *Chalasani et al., 2007*). We discovered that AMPK modulates AIB and AIY activity through two distinct mechanisms to ultimately ensure that adaptive foraging behaviours are appropriately triggered during periods of starvation. In the AIB interneurons AMPK regulates the abundance of the α-amino-3-hydroxy-5-methyl-4-isoxazolepropionic acid (AMPA) receptor GLR-1 in the postsynaptic elements, presumably by phosphorylation of serine 907 and 924 resulting in changes in synaptic strength and specific behavioural outputs. In addition, we also demonstrate that AMPK modulates a key metabotropic glutamate receptor called MGL-1 in the AIY interneuron at both mRNA and protein levels leading to an increase in AIY neuronal activity in starved animals. Together, our results indicate that AMPK acts as a starvation-inducible molecular trigger in the nervous system that modulates glutamatergic neuronal activity by at least two distinct mechanisms to modify behavioural outcomes in response to energy stress.

## Results

### Neuronal AMPK signalling triggers distal exploratory behaviour in starved animals

AMPK has been implicated in the regulation of feeding behaviours in higher animals (*Minokoshi et al., 2004*; *Yang et al., 2011*) although its essential role in development and cellular homeostasis has made it very difficult to study its role outside this closed circuit. Accordingly, the identification of the neuronal targets of this protein kinase that may be important for these behaviours has been virtually impossible to interrogate. Fortunately, *C. elegans* mutants that completely lack AMPK signalling are viable, thus providing us with an opportunity to investigate how AMPK regulates characteristic foraging behaviours in response to acute periods of starvation.

We and others found that removal of *aak-2*, the more prominent of two catalytic subunits of AMPK present in *C. elegans*, disrupts most cellular AMPK signalling and affects various aspects of *C. elegans* foraging behaviour (*Lee et al., 2008*; *Cunningham et al., 2014*). More specifically, *aak-2* mutants exhibit a behavioural profile in response to acute starvation that is more typical of satiated animals, where animals tend to explore locally rather than foraging in more distant locations. This is reflected by the frequency of body bends; which is considered as a representative readout for locomotory movement away from a nutrient-depleted environment in search of new resources, and reversals; a behaviour most often associated with local exploration (*Gray et al., 2005*). Both of these behaviours are affected in starved *aak-2* mutants (*Figure 1A*). These behaviours appear to be regulated predominantly by *aak-2* activity since removal of both catalytic subunits and hence eliminating all AMPK signalling (*aak(0)*) only modestly enhanced the phenotype of *aak-2* mutants in all our behavioural assays suggesting that *aak-2* mutants recapitulate a severe loss of AMPK function.

Since *aak-2* is a major regulator of foraging behaviour in *C. elegans*, we transferred mid-fourth larval stage (L4) *aak-2* mutant larvae to bacteria-free plates and monitored their movements for 20 hr to delineate sub-behaviours underlying *aak-2* deficiency. We found that starved *aak-2* mutants show defects in their rate of forward locomotion measured in the absence of food, which is compounded by their inability to appropriately suppress reversal behaviour during periods of more prolonged starvation (*Figure 1B*). Because we observe changes in each of these AMPK-dependent parameters in starved animals, and given the antagonistic relationship between forward and backward locomotion where increased reversal frequency affects the onset and/or duration of forward movement (*Burbea et al., 2002*; *Juo et al., 2007*), we chose to assess both behaviours in starved animals throughout our study.

The *aak-2*-dependent foraging defects we observed in starved animals are not likely to be due to a more general role of AMPK in the regulation of appropriate motor neuron development or function since AMPK mutants were comparable to wild type controls when reversal frequency and forward locomotion were quantified during conditions when food was abundant (*Figure 1—figure*

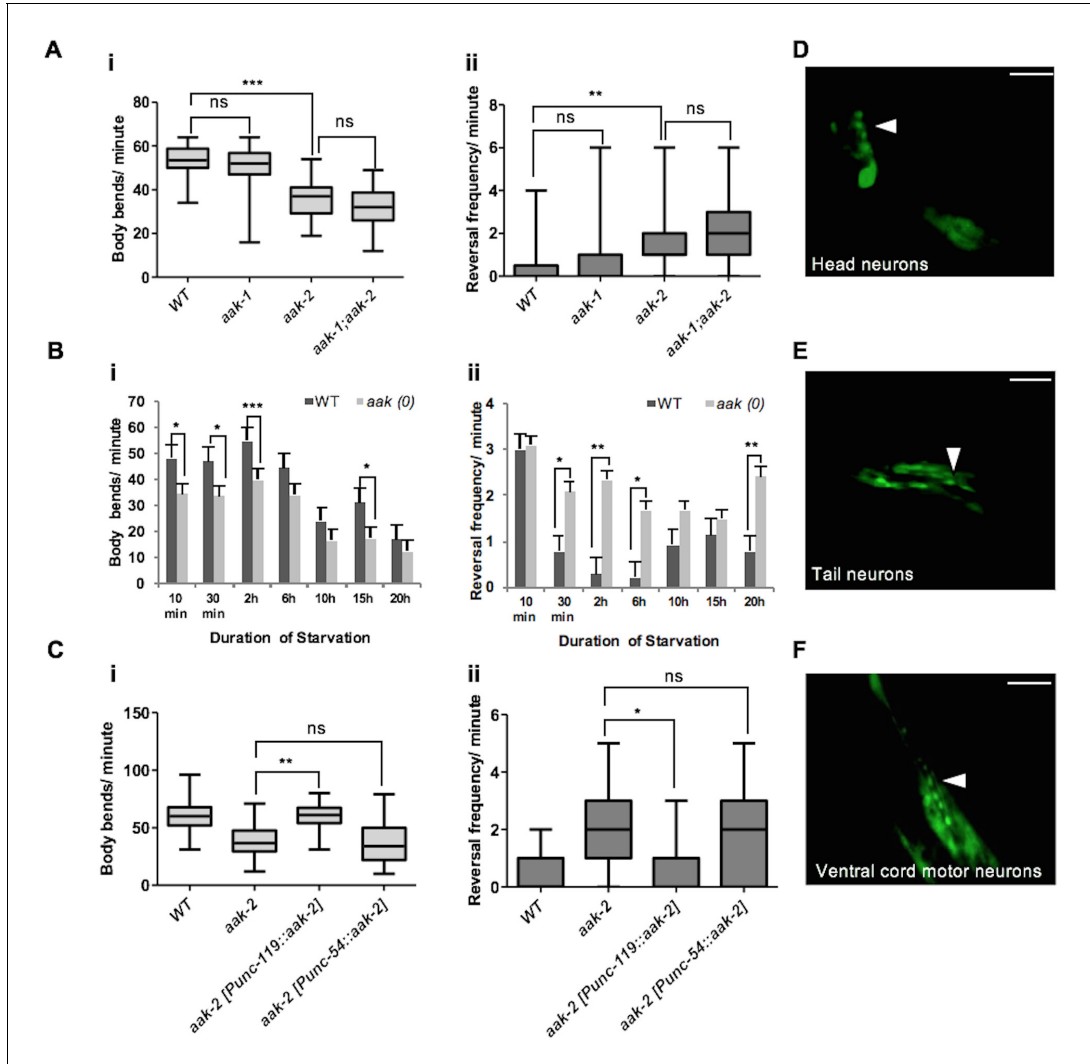

**Figure 1.** Neuronal AMPK regulates distal exploratory behaviour in starved animals. (**A**) Starved *aak-2* mutants display defective transition from local to distal exploration indicated by (i) decreased forward locomotion and (ii) increased reversal rate during acute bouts of starvation. These phenotypes are not dependent on *aak-1* (n>40), (one-way ANOVA **p<0.001, ***p<0.0001). (**B**) AMPK mutants display persistent decreased forward locomotion rate (i) and increased reversal frequency (ii) during periods of acute starvation (n>10), Error bars represent ± SEM (Student's t-test, *p<0.05, **p<0.001, ***p<0.0001). (**C**) *aak-2* reconstitution within the nervous system using the pan-neural [P*unc-119::aak-2*] transgene rescues the defective distal exploratory behaviour in starved *aak-2* mutants while its reconstitution within the body wall muscle using [P*unc-54::aak-2*] transgene does not improve the distal exploratory behaviour of *aak-2* mutants (n>15), (one-way ANOVA *p<0.05, **p<0.001). (**D,E,F**) *aak-2* is highly expressed throughout the nervous system. Scale bars are 40 μm. In the box and whisker plots (**A,C**) the central line is the median, the edges of the box are the 25th and 75th percentiles, and the whiskers extend to the most extreme data points.

The following source data and figure supplements are available for figure 1:

**Source data 1.** Locomotory behaviour in AMPK mutant animals.

**Source data 2.** Food and stimuli-related behaviours in AMPK mutants.

**Source data 3.** Locomotory behaviour upon depletion of aak-2 with 1535 RNAi.

**Figure supplement 1.** AMPK function is not required for other food-related behaviours rather than distal exploratory behaviour.

**Figure supplement 2.** Reduced forward locomotion rate in *aak-2* mutants is not a consequence of developmental defect in their nervous system.

*supplement 1A*). Moreover, *aak-2* mutants displayed a normal fleeing speed in response to a mechanical stimulus that was applied to the posterior (*Figure 1—figure supplement 1D*). In addition, we observed that *aak-2* (RNAi) performed at the late L2 stage, when the majority of neurons have already been generated, resulted in a reduction in forward locomotion compounded with an increase in reversal frequency in starved animals. This suggests that the reduced locomotory speed is unlikely to be the consequence of a general defect in motor function or neuromuscular development (*Figure 1—figure supplement 2*).

To discern whether *aak-2* is specifically required in neurons, muscle, or both tissues to modulate distal exploratory behaviours we used tissue-specific promoters to drive *aak-2* expression and test its ability to correct the behavioural defects typical of AMPK mutants. While an *aak-2* cDNA (*Narbonne and Roy, 2009*) expressed under the control of a pan neuronal promoter [P*unc-119::aak-2*] rescued the distal exploratory defect of *aak-2* mutants, *aak-2* mutants continued to exhibit prolonged local exploration when the same *aak-2* cDNA was reconstituted in body wall muscle using [P*unc-54::aak-2*] transgene (*Figure 1C*). Consistent with these results, we observed that a rescuing translational fusion reporter [P*aak-2::aak-2*::GFP] was broadly expressed throughout the nervous system (*Figure 1D,E,F*). These data are consistent with previous observations indicating that AMPK is required in the nervous system to regulate locomotory behaviour in response to food availability (*Lee et al., 2008*; *Cunningham et al., 2014*).

## AMPK is not required for all starvation-related behaviours

The ability of AMPK mutants to appropriately respond to starvation could result from a global AMPK-dependent defect in their nervous system rendering them incapable of responding to multiple cues in addition to starvation. However, this seems unlikely since two other food-related behaviours namely, basal slowing response and enhanced slowing response, which are triggered in well-fed and starved animals that are reintroduced to food, respectively, remain unaffected in AMPK mutants. In its natural habitat, a well-fed animal would be more likely to risk exploring distant locations for high quality food sources, whereas a starved animal would be less likely to stray far from a recently discovered vital food supply. These paradigms regulate behavioural plasticity in response to starvation in *C. elegans* and are mediated by distinct dopaminegic and serotonergic signalling (*Sawin et al., 2000*). When we re-introduced well-fed or starved *aak-2* mutants to food we did not detect any significant difference in either the basal slowing response or the enhanced slowing response (*Figure 1—figure supplement 1B,C*) suggesting that AMPK is not involved in the behavioural plasticity that occurs after re-introduction of animals to food and that the neural circuitry that mediates such behavioural plasticity is presumably intact and functional in animals that lack AMPK.

## AAK-2 is required in both the AIY and the AIB interneurons to mediate the transition from local to distal exploration in response to starvation

Since pan neuronal *aak-2* expression rescued the distal exploration defect of *aak-2* mutants, we next sought to identify the individual neurons that require *aak-2* activity to trigger this behaviour. The neural circuitry that dictates these simple behavioural outcomes has been well described (*White et al., 1976*, *1986*; *Chalfie et al., 1985*; *Wicks et al., 1996*; *Gray et al., 2005*). In particular, five pairs of command interneurons are required for the control of coordinated movement. The PVC and AVB interneurons are primarily required for the initiation of forward movement, while the AVA, AVD, and AVE interneurons control reversals (*White et al., 1976*; *Chalfie et al., 1985*; *Wicks et al., 1996*; *Gray et al., 2005*). Sensory inputs and serotonergic signalling together contribute to the behavioural changes that occur following removal from food (*Gray et al., 2005*). Moreover, several studies have underscored a critical role for both the AIY and the AIB interneurons, which are independently required to suppress and enhance reversals, respectively, resulting in longer or shorter durations of forward movement (*Tsalik and Hobert, 2003*; *Wakabayashi et al., 2004*; *Gray et al., 2005*; *Chalasani et al., 2007*; *Luo et al., 2014*). The RIM interneurons have also been determined as regulators of reversal frequency (*Gray et al., 2005*; *Gordus et al., 2015*).

To map the precise sites of AMPK function in its regulation of distal exploratory behaviour, we introduced *aak-2* in different subsets of neurons using neuronal sub-type-specific promoters: *tph-1* (serotonergic neurons) (*Cunningham et al., 2012*), *glr-1* (command interneurons) (*Zheng et al., 1999*), *che-2* (chemosensory neurons) (*Gray et al., 2005*), *ttx-3* (AIY) (*Chalasani et al., 2007*), *npr-9*

(AIB) (*Piggott et al., 2011*), *tdc-1* (RIM) (*Cunningham et al., 2012*) and *rig-3* (AVA) (*Marvin et al., 2013*). Reconstitution of AMPK (*aak-2*) using the *glr-1, tdc-1, ttx-3, npr-9* promoters was sufficient to partially rescue the defective distal exploration typical of *aak-2* mutants suggesting that AMPK functions within these neurons to phosphorylate targets involved in regulating these behavioural responses. Conversely, introducing AMPK (*aak-2*) in the sensory neurons, the serotonergic neurons, or the command interneurons involved in triggering reversals failed to correct the defect (*Figure 2A*). As AMPK expression within the GLR-1-expressing neurons partially rescued the locomotory defect of *aak-2* mutants, it is very likely that AMPK may be independently required in the GLR-1-expressing neurons AIB and RIM, although this does not exclude a role for *aak-2* in other GLR-1-expressing neurons.

The AIY and AIB interneurons receive synapses from the AWC sensory neurons and act in parallel to suppress and enhance reversals, respectively, in response to food availability (*Figure 2—figure supplement 1A,B*) (*Chalasani et al., 2007*). The AIB interneurons relay their signals to the RIM and AVA, which in turn send their outputs to the muscles (*Figure 2—figure supplement 1C*) (*Gordus et al., 2015*). As an AMPK expression within the AIY, AIB and RIM partially restored the defective distal exploration of starved *aak-2* mutants, it is likely that AMPK functions in multiple neurons in a parallel circuit to regulate locomotory behaviour in response to starvation.

When we expressed *aak-2* in both the AIY and the AIB interneurons, we completely restored the defective distal exploration of *aak-2* mutants, while expression of *aak-2* within the RIM and the AIB interneurons, or in RIM and AIY, did not improve the distal exploration defect of starved *aak-2* mutants beyond the rescue observed in AMPK mutant animals expressing *aak-2* within the AIB or the AIY interneurons alone (*Figure 2B*). The RIM neurons form chemical and electrical synapses with AIB and AIY, respectively (*White et al., 1986*). Of note, it has been shown that chemical synapses from the RIM neurons enhance the variability in response to odour in the AWC-AIB-RIM olfactory circuit (*Gordus et al., 2015*). Therefore, *aak-2* expression within the RIM neuron might influence this variability, whereby the expression of *aak-2* within the AIB and RIM neurons biases the circuit towards either a more probabilistic or reliable output. Taken together, our data highlight a critical role of AMPK in the AIY and AIB interneurons to regulate the appropriate transition from local to distal exploration in animals subjected to acute starvation.

## AMPK functions within the AIB and AIY interneurons to integrate sensory signals in response to starvation

AMPK mutants demonstrate reduced forward movement accompanied by an increased frequency of reversals. If these defects disrupt distal exploratory foraging behaviour then starved *aak-2* mutants could also be defective in finding food at a distance. We therefore placed starved animals on plates with a point food source (*E. coli)* and monitored the time required for each animal to encounter the bacteria. We noted that *aak-2* mutants were substantially less efficient than wild type controls in their ability to track to the food source (*Figure 2C*). However, from this assay alone, we were unable to distinguish if this defect is a consequence of a deficiency in their ability to sense food or to integrate the detection of nutritional signals to appropriately modify locomotory behaviour since we also observed a significant defect in the ability of AMPK mutants to chemotaxis toward the attractant iso-amyl alcohol (IAA) (*Figure 2D*). However, since the expression of *aak-2* within the sensory neurons failed to restore the defective distal exploratory behaviour in *aak-2* mutants, while this defect was fully rescued by driving *aak-2* expression specifically within the AIB and AIY interneurons, we conclude that AMPK is critical for the integration of sensory cues and not with chemosensation per se (*Figure 2D*). Taken together, our results indicate that AMPK is required in the AIB and AIY interneurons to trigger appropriate exploratory behaviour in response to acute starvation.

## AMPK functions upstream of *mgl-1*, *glr-1* and *eat-4* to modulate distal exploration

To position how and where AMPK functions within the AIY and AIB interneurons to modulate the transition between local and distal exploration, we performed optogenetic experiments to manipulate AIB and AIY activity in starved *aak-2* mutants (*Kocabas et al., 2012*). ChR2-mediated depolarization of the AIY interneurons in *aak-2* mutant animals resulted in a significant decrease in reversal frequency that was persistent throughout the illumination time. In contrast, AIY activation in WT

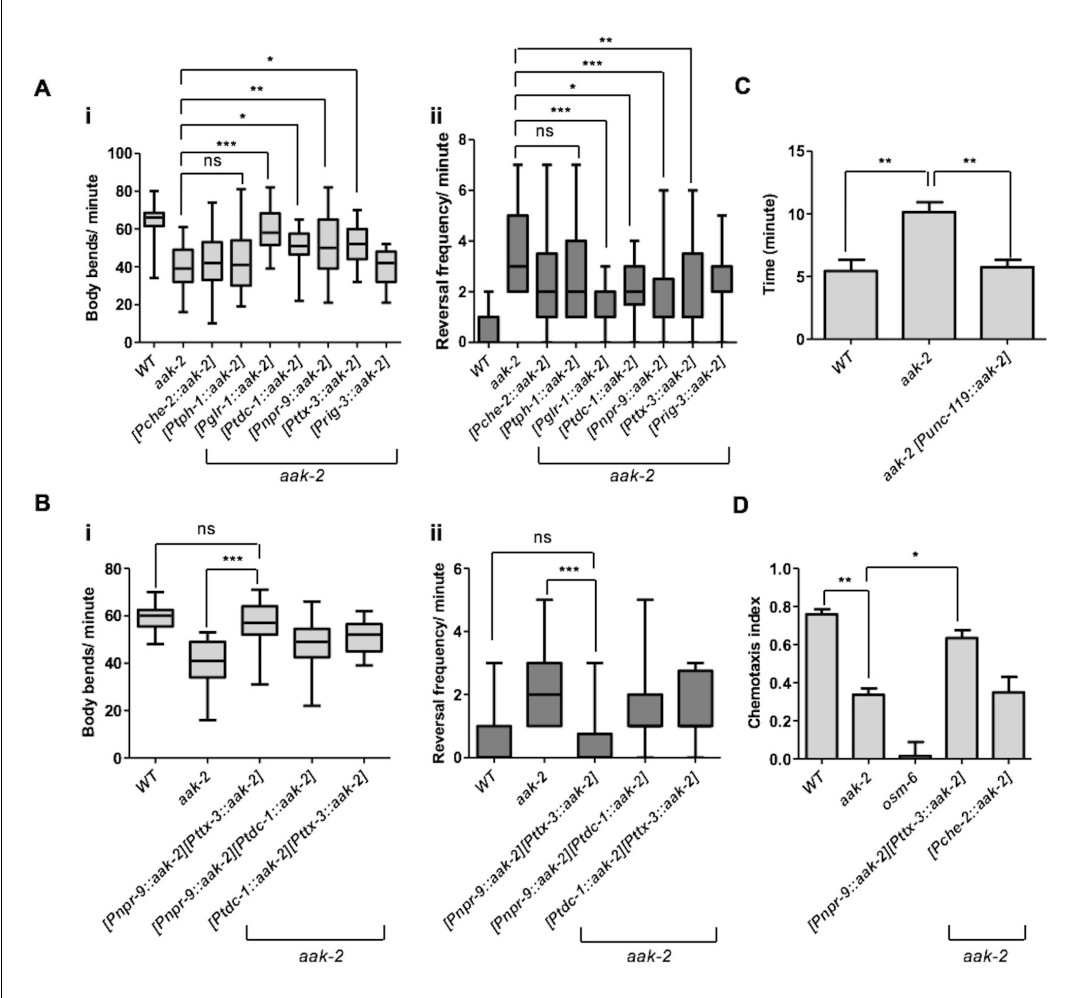

**Figure 2.** AMPK acts in the AIB and AIY interneurons to integrate chemosensory signals and trigger distal exploratory behaviour in starved animals. (**A**) Targeted expression of *aak-2* within the GLR-1-expressing neurons [P*glr-1::aak-2*], RIM neurons [P*tdc-1::aak-2*], AIB interneurons [P*npr-9::aak-2*] and AIY interneurons [P*ttx-3::aak-2*], but not chemosensory neurons [P*che-2::aak-2*], serotonergic neurons [P*tph-1::aak-2*] or AVA neurons [P*rig-3::aak-2*] partially restores the distal exploratory defect in starved *aak-2* mutants by affecting both forward locomotion (i) and reversal frequency (ii) (n>20), (one-way ANOVA *p<0.05, **p<0.001, ***p<0.0001). (**B**) Simultaneous rescue of *aak-2* in the AIB and AIY using [P*npr-9::aak-2*][P*ttx-3::aak-2*] transgenes is sufficient to completely restore the defective exploratory behaviour typical of starved *aak-2* mutants (n>20), (one-way ANOVA ***p<0.0001). (**C**) Starved worms were placed 1.5 cm away from a spot of food (fresh OP50) and the time required to encounter the food was monitored for the worms that found food within 16 minutes. *aak-2* mutants display defective food detection indicated by increased time spent to find food and this defect can be rescued by specific expression of *aak-2* throughout the nervous system using [P*unc-119::aak-2*] transgene (n>10). (**D**) Targeted expression of *aak-2* within the AIB and AIY interneurons, but not the chemosensory neurons rescue the defective chemosensation of *aak-2* mutants toward IAA (n>300). Error bars in (**C,D**) represent ± SEM (one-way ANOVA *p<0.05, **p<0.001).

The following source data and figure supplement are available for figure 2:

**Source data 1.** AMPK reconstitution in different neurons to restore the defective distal exploratory behaviour and defective chemotaxis typical of aak-2 mutants.

**Figure supplement 1.** Neural circuits engaged in the regulation of locomotory behaviour in *C. elegans*.

animals slightly altered the forward locomotion and reversal rate which is consistent with previous studies demonstrating the activation of the AIY interneurons in starved animals (*Gray et al., 2005*; *Flavell et al., 2013*) (*Figure 3A*, *Figure 4—figure supplement 2A*).

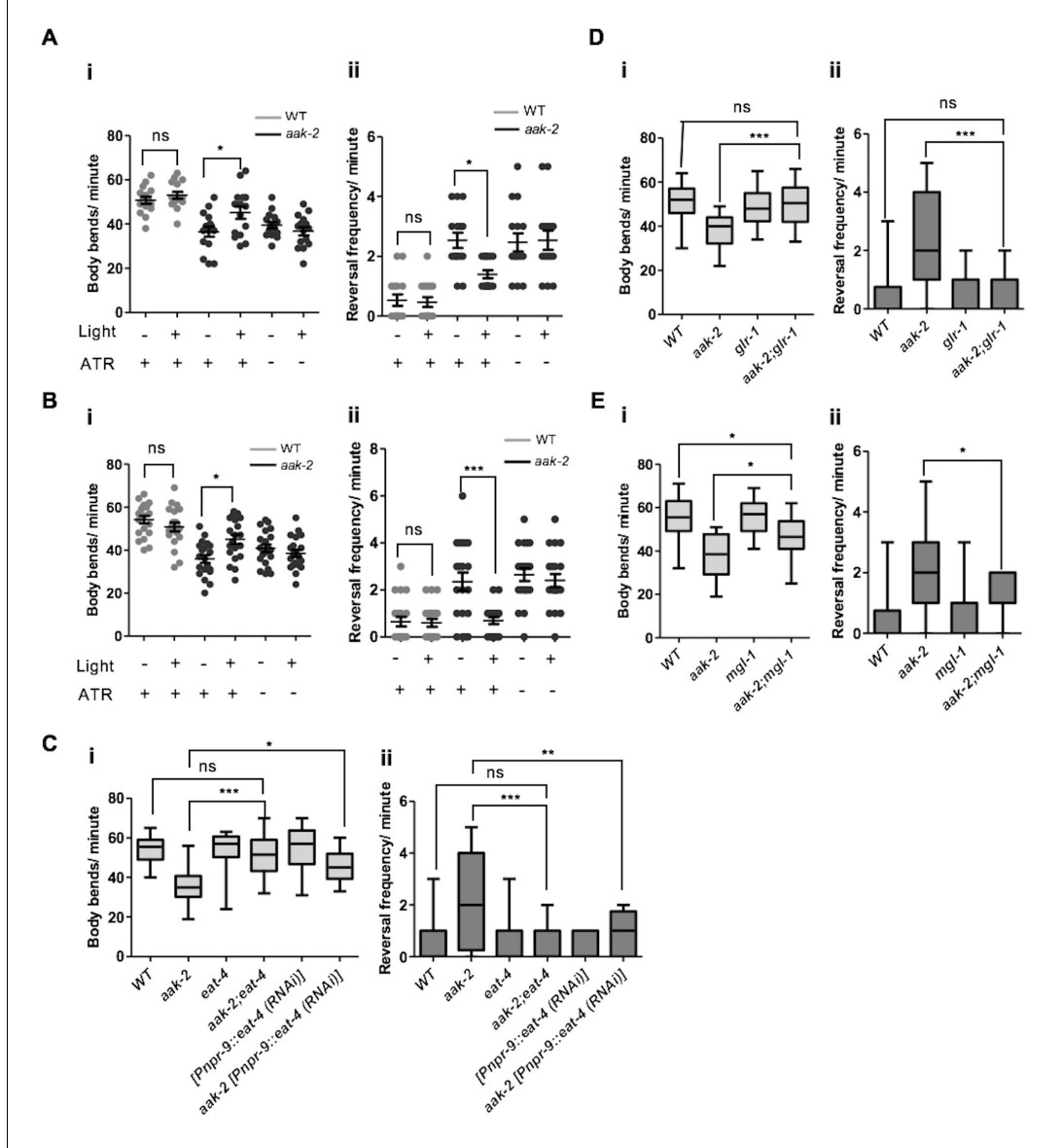

**Figure 3.** AMPK regulates glutamatergic signalling. (**A,B**) ChR2-mediated activation of AIY interneurons (**A**), or ARCH -mediated silencing of AIB interneurons (**B**) in starved *aak-2* mutants grown in the presence of all-trans retinal (ATR) induces distal exploration by suppressing reversals and inducing forward locomotion (n>15), Error bars represent ± SEM (two-way ANOVA *p<0.05, ***p<0.0001). (**C,D,E**) Compromised *eat-4, glr-1* or *mgl-1* function rescues the defective exploratory behaviour of starved *aak-2* mutants by inducing forward locomotion (**i**) and suppressing reversals (**ii**) (n>20), (one-way ANOVA *p<0.05, **p<0.001, ***p<0.0001). In the box and whisker plots (**C,D,E**) the central line is the median, the edges of the box are the 25th and 75th percentiles, and the whiskers extend to the most extreme data points.

The following source data and figure supplement are available for figure 3:

**Source data 1.** Optogenetics and epistatic analysis.

**Source data 2.** The analysis of locomotory behaviour of potential AMPK phospho targets.

**Source data 3.** Potential AMPK phosphorylation targets expressed within the AIB and AIY interneurons.

**Figure supplement 1.** *glr-1, eat-4* and *mgl-1* are epistatic to *aak-2*.

In a complementary experiment, we assessed the behaviour of starved *aak-2* mutants following inactivation of AIB induced by the light-gated chloride pump archaerhodopsin-3 (Arch) (*Kocabas et al., 2012*). The Arch-induced hyperpolarization of AIB was sufficient to suppress reversals in the *aak-2* mutants, while also promoting normal distal exploratory behaviour during starvation. In contrast, Arch-induced AIB inactivation in WT animals altered their locomotory behaviour slightly which can be explained by inactivation of AIB during acute bouts of starvation as the locomotory behaviour of starved animals has been shown to remain unchanged upon AIB ablation (*Gray et al., 2005*) (*Figure 3B*). We interpret these observations to indicate that although AMPK activity is required within the AIB and AIY interneurons for appropriate distal exploration in response to acute starvation, it does so without negatively impacting normal neurotransmission, neuronal development or connectivity within the interneurons.

Having identified the interneurons in which *aak-2* is required to suppress reversals, promote forward locomotion, and eventually trigger distal exploration, we then set out to identify AMPK targets within these interneurons that mediate its effects in response to starvation. Using bioinformatic tools we scanned the *C. elegans* proteome for proteins that possessed consensus AMPK phosphorylation sites and were known to be expressed in the AIB or AIY interneurons, providing us with a list of potential neuronal phosphorylation targets of AMPK (*Figure 3—source data 3*).

Among the candidates that we identified through our bioinformatic analysis GLR-1, EAT-4, and MGL-1 emerged as highly predicted phosphorylation targets of AMPK (*Figure 3C,D,E*, *Figure 3— figure supplement 1A*). *eat-4* encodes an ortholog of a mammalian brain-specific sodium-dependent inorganic phosphate co-transporter I (BNPI), and is required for glutamatergic neurotransmission in the AIB (*Lee et al., 1999*; *Piggott et al., 2011*). Interestingly, we found that *eat-4* is epistatic to AMPK function during the response to acute starvation; single *eat-4* mutants behave like starved WT animals by decreasing reversal frequency in contrast to the *aak-2* mutants that exhibit an increased number of reversals. The *aak-2; eat-4* double mutants behave more like starved WT (or *eat-4*) animals than *aak-2* mutants (*Figure 3C*) indicating that *eat-4* acts downstream of, or in parallel to AMPK, in the genetic pathway controlling this starvation-induced behaviour.

The AMPA-type GluR GLR-1 is expressed in both the command interneurons and in some motor neurons, and is implicated in memory formation and the behavioural responses to light nose touch and to sensory cues such as food (*Hart et al., 1995*; *Rose et al., 2003*; *Chalasani et al., 2007*). Moreover, GLR-1 function is required for glutamatergic activity in the AIB (*Chalasani et al., 2007*). Similar to *eat-4*, mutations in *glr-1* suppress the defective distal exploratory behaviour of AMPK mutants suggesting that like *eat-4*, *glr-1* is epistatic to *aak-2* (*Figure 3D*). Taken together, these results indicate that blocking the glutamatergic inputs to the AIB through the elimination of *glr-1*, or blocking its glutamatergic neurotransmission by disrupting *eat-4*, can compensate for the behavioural defects in starved *aak-2* mutants. Our data are therefore consistent with *aak-2* acting upstream of, or in parallel to these two glutamatergic effectors to modulate neuronal activity in the AIB interneuron during periods of acute starvation.

As initially determined in mammals, the metabotropic glutamate receptors (mGluRs) also have an important neuromodulatory role in glutamatergic transmission within the *C. elegans* nervous system (*Dillon et al., 2006*, *2015*). *mgl-1* encodes one of the three metabotropic glutamate receptors in *C. elegans* and is expressed throughout the AIY interneurons. Based on sequence analysis, MGL-1 is a group II metabotropic glutamate receptor that acts as both pre- and postsynaptic detectors of glutamate to reduce neuronal excitation at least partly by inhibition of adenylyl cyclase activity (*Dillon et al., 2015*; *Niswender and Conn, 2010*). Previous studies have identified *mgl-1* as an essential component for the appropriate regulation of fat accumulation (*Greer et al., 2008*) and the starvation responses mediated by AIY (*Kang and Avery, 2009*). Our genetic analyses indicate that mutations in *mgl-1* could also significantly, but not completely, suppress the prolonged local exploration typical of *aak-2* mutants whereby a compromise of *mgl-1* in the *aak-2* background results in reduced reversals and increased distal runs (*Figure 3E*). These results suggest that AMPK may negatively regulate MGL-1 activity in the AIY in response to starvation.

Our data are consistent with AMPK acting as a potential modulator of glutamatergic inputs in both the AIB and the AIY, but it cannot distinguish whether AMPK affects glutamatergic inputs in a subset of glutamatergic neurons (i.e.,the AIB and AIY) or in all glutamatergic neurons. Furthermore, if AMPK does have a specific role in the AIB and the AIY, then what determines its functionality in this subset of AMPK-sensitive neurons? To test if AMPK affects other mediators of glutamatergic

inputs we assessed whether AMPK compromise could modify the phenotypes associated with: *nmr-1*, which encodes a NMDA-type ionotropic glutamate receptor subunit; the glutamate-gated chloride channels *avr-14* and *mgl-2* which encodes a group I metabotropic glutamate receptor (*Chalasani et al., 2007*; *Piggott et al., 2011*; *Dillon et al., 2015*). Double mutant combinations with *aak-2* did not modify any of the *aak-2* phenotypes we tested (*Figure 3—figure supplement 1B*) suggesting that *aak-2* does not affect glutamatergic transmission generally, but rather it seems to only affect a subset of glutamatergic neurons critical for starvation-dependent functions and/or behaviours.

## Parallel and opposing function of AMPK in the modulation of AIB and AIY outputs

Given the epistatic relationship between *aak-2* and genes involved in glutamatergic signalling we wanted to clarify if AMPK regulates the AIB and AIY neural activity through these targets. We monitored calcium transients using the genetically-encoded calcium sensor G-CaMP individually in both interneurons as a proxy for neuronal activity (*Figure 4A,B,C,D,E,F*). We observed a decrease in AIY activity in starved *aak-2* mutants that was rescued by AIY-specific *aak-2* expression, consistent with *aak-2* regulating AIY activity/response during starvation (*Figure 4A,B,C,G*, *Figure 4—figure supplement 1A*). Previous studies have shown that calcium spikes in the AIY interneurons correlate positively with forward runs and negatively with the initiation of reversals with gradual increases preceding forward run initiation (*Flavell et al., 2013*; *Luo et al., 2014*). Our results suggest that the decreased AIY activity in starved *aak-2* mutants at least partially contributes to their defective distal exploratory behaviour. To further investigate whether the increased reversal rate correlates with decreased AIY activity in starved *aak-2* mutants, we monitored the calcium levels in the AIY interneuron of freely behaving animals. The AIY calcium levels were consistently higher upon the termination of reversals, and during the subsequent forward runs with gradual increase preceding forward run initiation suggesting that the calcium peaks in the AIY correlate with the suppression of reversals and the observed acute increase in forward speed (*Figure 4—figure supplement 2A*). Furthermore, consistent with our previous results, we found that starved *aak-2* mutants display increased reversal frequency and shorter durations of forward locomotion, both of which correlated with decreased calcium influx and associated AIY activity (*Figure 4—figure supplement 2C,E,F*).

The epistatic relationship between *mgl-1* and *aak-2* in distal exploration was further corroborated by the total integrated fluorescence intensity demonstrated by *aak-2; mgl-1* animals. These readings resemble both starved WT and/or *mgl-1* mutants, reinforcing our data indicating that *mgl-1* is epistatic to *aak-2* function in the AIY interneuron during acute starvation, where MGL-1 potentially acts as a postsynaptic receptor in the AIY (*Figure 4H—figure supplement 1A*). AIY-ablated animals are defective in transitioning between local and distal exploration (*Gray et al., 2005*), these results reveal an additional, albeit critical, role of AMPK within the AIY interneurons to modulate AIY neuronal activity autonomously, thus contributing to the behavioural transition associated with the decision to explore more distal environments when animals are subjected to starvation.

Unlike the AIY interneurons, activity in AIB increased in starved *aak-2* mutants and could be rescued by AIB-specific *aak-2* expression (*Figure 4D,E,F,I*). Calcium peaks in the AIB have been shown to correlate with reversal frequency (*Piggott et al., 2011*; *Gordus et al., 2015*), but during our recording the animals are restrained therefore we presume that each calcium peak is representative of a reversal attempt. Given the increased calcium levels in the AIB interneurons in starved *aak-2* mutants, our data indirectly suggest that the increased reversal frequency and the abnormally prolonged local exploration is at least partly a consequence of increased AIB neuronal activity. To further examine if the increased AIB activity in starved *aak-2* mutants correlates with their increased reversal frequency, we examined the behaviour of freely moving animals during acute starvation. The calcium peaks in the AIB were well correlated with reversal frequency, further suggesting that the increased reversal frequency in starved *aak-2* mutants is at least partially due to the increased AIB activity (*Figure 4—figure supplement 2B,D,G,H*).

The genetic requirement for *aak-2* in the AIB interneurons to modulate neuronal activity, along with its epistatic relationship with *eat-4* and *glr-1* in the modulation of locomotory behaviour, are consistent with at least two models: During acute starvation *aak-2* might regulate excitatory synaptic inputs onto the AIB interneurons through modulation of *glr-1* activity, or *aak-2* might affect glutamatergic neurotransmission from the AIB interneurons through, or in parallel with *eat-4*. To distinguish

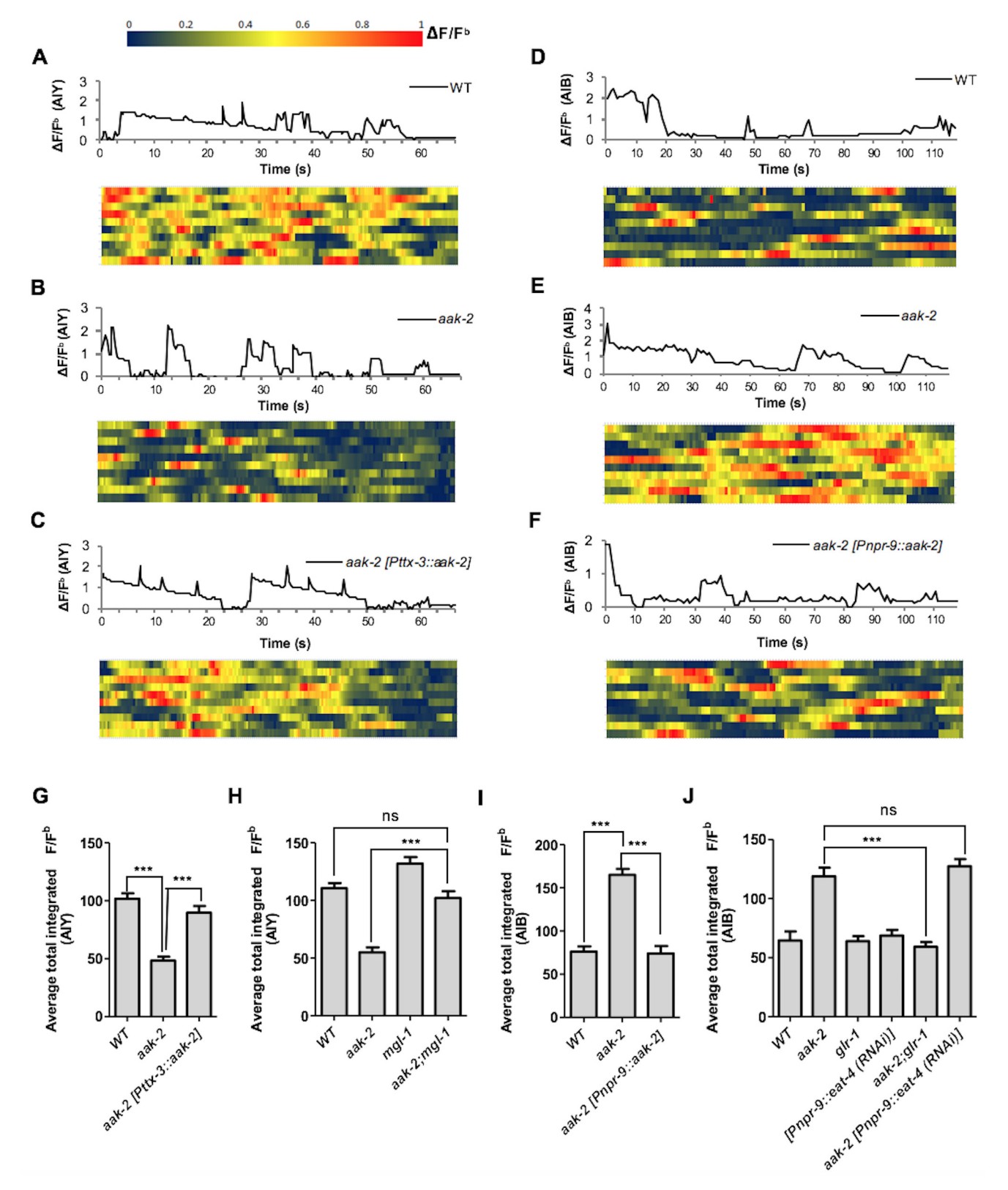

**Figure 4.** AMPK regulates AIB and AIY neuronal activity. (**A,B,C,D,E,F**) A sample $\Delta F/F^b$ plot showing spontaneous changes in the AIY and AIB neuronal activity in starved individuals during the course of a 65 and 120 s imaging window, respectively. These plots are the basis of the data in **G**, **H**, **I** and **J**. The heatmaps show normalized G-CaMP1 and G-CaMP3 traces in AIY and AIB, respectively in multiple animals. (**G**) The average total integrated

*Figure 4 continued on next page*

*Figure 4 continued*

fluorescence intensity (ΔF/F$^b$) over the course of a 65 s window. AIY neuronal activity is reduced in starved *aak-2* mutants and this reduction in the AIY neuronal activity is restored by targeted expression of *aak-2* within the AIY interneurons using the AIY-specific [P*ttx-3::aak-2*] transgene (n>10). (**H**) Removal of *mgl-1* in *aak-2* mutants rescues the reduced AIY neuronal activity observed in starved *aak-2* mutants (n>10). (**I**) The average total integrated fluorescence intensity (ΔF/F$^b$) in AIB is increased upon removal of *aak-2* in starved animals, and this increase is reversed by expression of *aak-2* within the AIB using the AIB-specific [P*npr-9::aak-2*] transgene (n>10). (**J**) Removal of *glr-1*, but not *eat-4*, restores the increased AIB neuronal activity in starved *aak-2* mutants (n>10). Error bars represent ± SEM (one-way ANOVA ***p<0.0001).

The following source data and figure supplements are available for figure 4:

**Source data 1.** Calcium imaging in the AIB and AIY interneurons in WT, aak-2, mgl-1, glr-1 and eat-4 and double mutants.
**Source data 2.** Calcium imaging in freely WT and aak-2 behaving animals.
**Figure supplement 1.** Heat maps showing AIY and AIB spontaneous neuronal activity in starved animals.
**Figure supplement 2.** AIY and AIB spontaneous neuronal activity in starved freely behaving animals.

between these possibilities, we monitored calcium transients in *aak-2; glr-1* and *aak-2;* [P*npr-9::eat-4* (RNAi)] double mutants. Of note, we observed that although blocking the AIB neurotransmission in *aak-2;* [P*npr-9::eat-4* (RNAi)] animals resulted in normal distal exploratory behaviour during starvation (*Figure 3C*), the total integrated signal was significantly higher compared to both WT and [P*npr-9:: eat-4* (RNAi)] animals (*Figure 4J*, *Figure 4—figure supplement 1B*). This suggests that *aak-2* acts independently of *eat-4* to modulate neural activity in the AIB.

Taken together, our genetic analyses indicate that AMPK is required within the AIY and the AIB interneurons to modulate their neuronal activity through its direct or indirect regulation of the glutamatergic receptors GLR-1 and MGL-1 during starvation-induced exploratory behaviour.

## AMPK controls GLR-1 abundance in the neuronal postsynaptic membranes

Increased GLR-1-mediated glutamatergic inputs have been shown to bias the locomotory circuit toward reversals (*Zheng et al., 1999*). Consistent with this, overexpression of GLR-1 specifically within the AIB interneurons increases reversal frequency and turning rate in the absence of food (*Chalasani et al., 2007*). Because the abundance of GLR-1 within the AIB influences reversal behaviour, we tested whether the increased reversal frequency in *aak-2* mutants is a consequence of changes in GLR-1 levels. We quantified *glr-1* mRNA levels in WT and *aak-2* mutants under well-fed and starved conditions and we did not detect any differences (*Figure 5—figure supplement 1A*). However, we did notice that total GLR-1 protein levels in starved WT animals were reduced and this effect was less pronounced in starved *aak-2* mutants (*Figure 5—figure supplement 1B*) suggesting that AMPK may modulate GLR-1 protein levels in starved animals either at the level of synthesis or stability.

Studies have demonstrated that disruption of GLR-1 endocytosis results in increased GLR-1 abundance in the postsynaptic elements which results in increased synaptic strength and reversal frequency (*Burbea et al., 2002*; *Juo et al., 2007*). Given the epistatic relationship between *aak-2* and *glr-1* we next asked if the increased reversal frequency in starved *aak-2* mutants occurs as a result of decreased GLR-1 turnover linked to a change in its post-translational regulation due to the absence of AMPK. To examine this possibility, we used a previously described chimaeric receptor tagged with the green fluorescent protein (GLR-1::GFP) that localizes to discrete punctate structures and that can be used to visualize central glutamatergic synapses in living animals (*Burbea et al., 2002*). We measured the density of GLR-1::GFP-containing puncta in the AIB neuronal process in a strain that expresses GLR-1::GFP driven by the AIB-specific promoter *npr-9*. Notably, we observed that the GLR-1 levels in the AIB neuronal process in the nerve ring where the AIB interneurons form their synapses (*Figure 5A*) was significantly greater in *aak-2* mutants than WT animals during starvation (*Figure 5B,C*). Moreover, this difference could be rescued by introducing AMPK specifically within

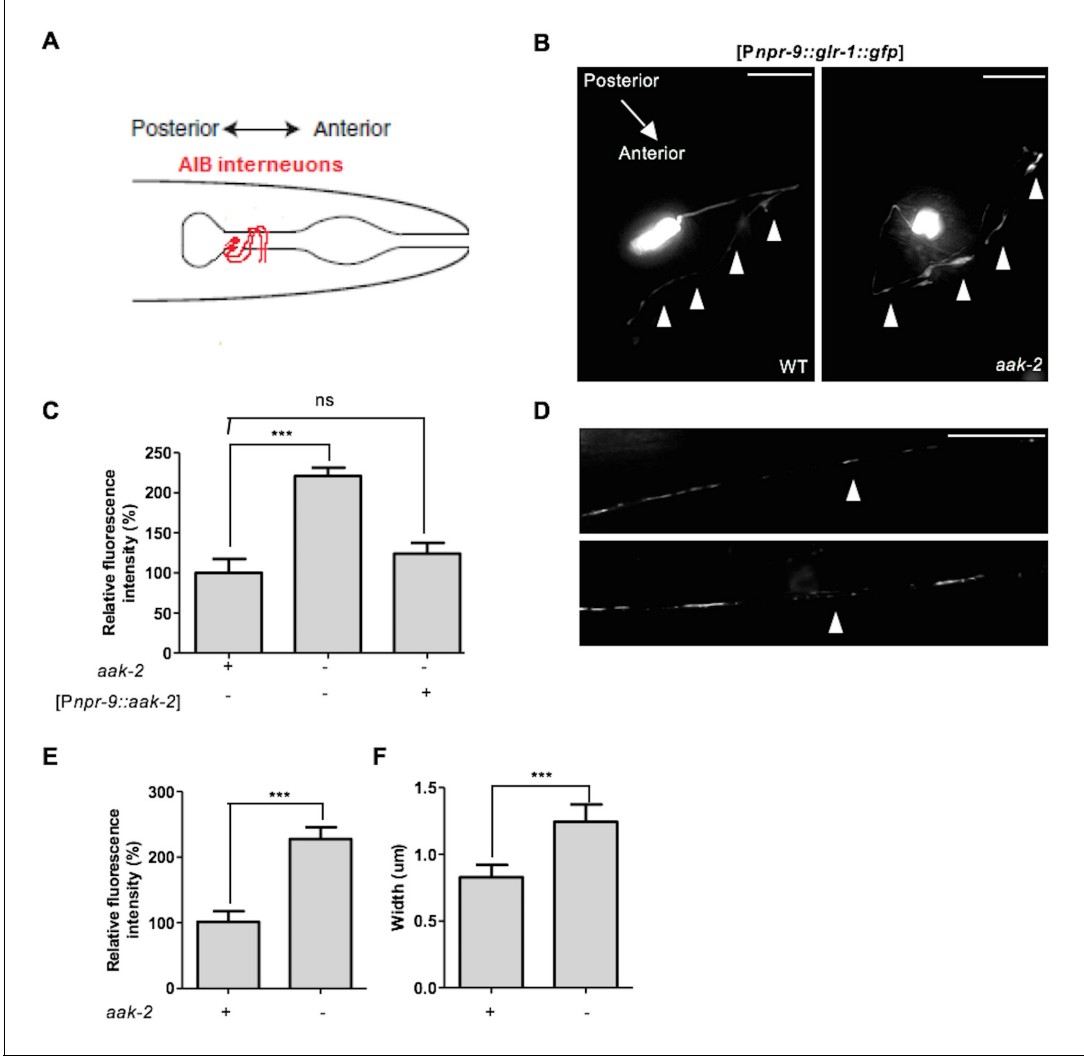

**Figure 5.** AMPK regulates GLR-1 abundance in the AIB neuronal process and the ventral nerve cord. (A) Schematic representation showing AIB interneurons that extend their processes into the nerve ring. (B,C) Representative images of [Pnpr-9::glr-1::gfp] in the AIB neuronal processes in starved WT and aak-2 mutants (B). GLR-1::GFP level is significantly increased in the AIB neuronal processes in the aak-2 mutants compared to WT animals under starvation condition. Targeted expression of aak-2 within the AIB interneurons rescues the increased GLR-1::GFP level in starved aak-2 mutants (C). At least 3 [Pnpr-9::glr-1::gfp] transgenic lines were separately examined for this experiment (n>20), Error bars represent ± SEM (one-way ANOVA ***p<0.0001). Scale bars are 10 μm. (D,E,F) Representative images of [Pglr-1::glr-1::gfp] in the ventral nerve cord (D). GLR-1::GFP level (E) and puncta width (F) are significantly increased in the ventral nerve cord of starved aak-2 mutants compared to WT animals. (n>20), Error bars represent ± SEM (Student's 2-tailed t test ***p<0.0001). Scale bars are 10 μm.

The following source data and figure supplement are available for figure 5:

**Source data 1.** Measurement of GLR-1 abundance in the AIB neuronal process and ventral nerve cord of WT and aak-2 mutants.
**Source data 2.** The extent of GLR-1 and EAT-4 colocalization in WT and aak-2 mutants.
**Figure supplement 1.** AMPK regulates GLR-1 abundance by affecting steady state protein levels.

the AIB interneurons (*Figure 5C*), highlighting the importance of AMPK in modulating GLR-1 abundance within the AIB interneurons.

Our data suggest that AMPK regulates the GLR-1 abundance in the AIB interneurons to potentially modulate synaptic inputs and consequently affect reversal behaviour. But aak-2 mutants have additional behavioural defects that may also manifest due to inappropriate glutamatergic signalling

(*Cunningham et al., 2012*). Therefore, to further investigate if the effect of AMPK on GLR-1 abundance is limited to the AIB interneurons or alternatively AMPK is globally required for the modulation of GLR-1 steady state levels throughout the nervous system, we measured the density of GLR-1::GFP-containing puncta in the anterior region of the ventral nerve cord between the RIG neuron cell bodies, and in the vulva using the [P*glr-1::glr-1::gfp*] transgene (*Burbea et al., 2002*). Interestingly, we observed a significant increase in the GLR-1 levels and the size (diameter) of GLR-1-expressing puncta in starved *aak-2* mutant animals compared to starved WT controls (*Figure 5D,E, F*). Furthermore, we noted that the GLR-1::GFP signal in *aak-2* mutants overlapped with that of EAT-4 which marks the majority of glutamatergic synapses (in *Figure 5—figure supplement 1C,D,E*). To summarize, the removal of *aak-2* results in an increase in GLR-1 abundance in starved animals and a large fraction of the GLR-1::GFP puncta seen in the mutants correspond to postsynaptic elements.

As AIB interneurons form their synapses in the nerve ring, the increased GLR-1 abundance in the ventral nerve cord of *aak-2* mutants must be a consequence of AMPK compromise in additional GLR-1-expressing neurons. Since the quantification of fluorescent intensity and size/width of the puncta are two determinants of total receptor abundance at each synapse, these results collectively suggest that GLR-1 levels at the postsynaptic elements are globally changed in *aak-2* mutants. Therefore, AMPK compromise is associated with increased steady state GLR-1 protein levels and GLR-1::GFP fluorescence, and that most of the GLR-1 receptors may be degraded in an AMPK-dependent manner, or alternatively, that GLR-1 synthesis may also be affected in starved *aak-2* mutants.

## AMPK acts with UNC-11/AP180 to regulate GLR-1 levels within the postsynaptic elements and consequently modulate synaptic strength

Mammalian AMPA receptors are removed from postsynaptic membranes by clathrin-mediated endocytosis (*Carroll et al., 1999*; *Man et al., 2000*). In *C. elegans*, the ubiquitylation of conserved cytoplasmic lysine residues on GLR-1 constitutes an endocytic signal to remove the receptors from postsynaptic elements which subsequently targets them for degradation; a process that is mediated by the *unc-11*/AP180 clathrin adaptin protein (*Burbea et al., 2002*). To further investigate if AMPK modulates GLR-1 abundance by affecting endocytosis in response to starvation, we measured GLR-1 abundance in *unc-11* mutants that are unable to internalize GLR-1. The GLR-1 levels in *aak-2; unc-11* double mutants were indistinguishable from those found in either *unc-11* single mutants (*Figure 6A, B,C*). Similarly, the increased GLR-1 levels and size/width of GLR-1 puncta measured in the *aak-2* mutants was not further enhanced when all four cytoplasmic lysine residues required for GLR-1 ubiquitylation and endocytosis were mutated to arginine [GLR-1(4KR)::GFP], further suggesting that *aak-2* acts in a linear pathway with an *unc-11*/clathrin-mediated mechanism to regulate GLR-1 endocytosis (*Figure 6A,B,C*).

Activation of AMPK during periods of starvation could result in the regulation of GLR-1 abundance by its direct phosphorylation through its consensus phosphorylation sites. To determine the importance of these sites we used the [P*npr-9::glr-1::gfp*] fusion protein as a template to generate a GLR-1::GFP variant with non-phosphorylable AMPK sites. The cytoplasmic domain of GluR has already been described as an important phosphoregulatory target, mostly by kinases such as PKA and PKC (*Roche et al., 1996*). Since two conserved AMPK phosphorylation sites were identified within the cytoplasmic domain of GLR-1 (*Figure 5—figure supplement 1F*), we mutated both S907 and S924 to non-phosphorylable alanine residues to evaluate how these sites contribute to the stability of GLR-1 within the AIB neuronal process during periods of starvation. Wild type animals bearing the compound (S907A, S924A) variant transgene demonstrated defective exploratory behaviour that was comparable to *aak-2* mutants (*Figure 6D*). To further examine if the GLR-1 receptors that accumulate in AMPK mutants act as functional synaptic receptors we examined the locomotory behaviour of *eat-4; glr-1* double mutants expressing the non-phosphorylable variant (S907A, S924A). We noticed that the increased reversal frequency rate accompanied by reduced forward locomotion in animals expressing the (S907A, S924A) variant was largely suppressed in the absence of *eat-4* indicating that the compromise of AMPK signalling results in an increased abundance of functional synaptic GLR-1 receptors (*Figure 6D*). Furthermore, the GLR-1 (S907A, S924A) variant accumulated in the AIB neuronal process and this accumulation was not enhanced upon removal of *unc-11* (*Figure 6E*), further suggesting that AMPK regulates GLR-1 abundance directly by phosphorylating the protein to mediate UNC-11/AP180-dependent GLR-1 endocytosis in response to starvation.

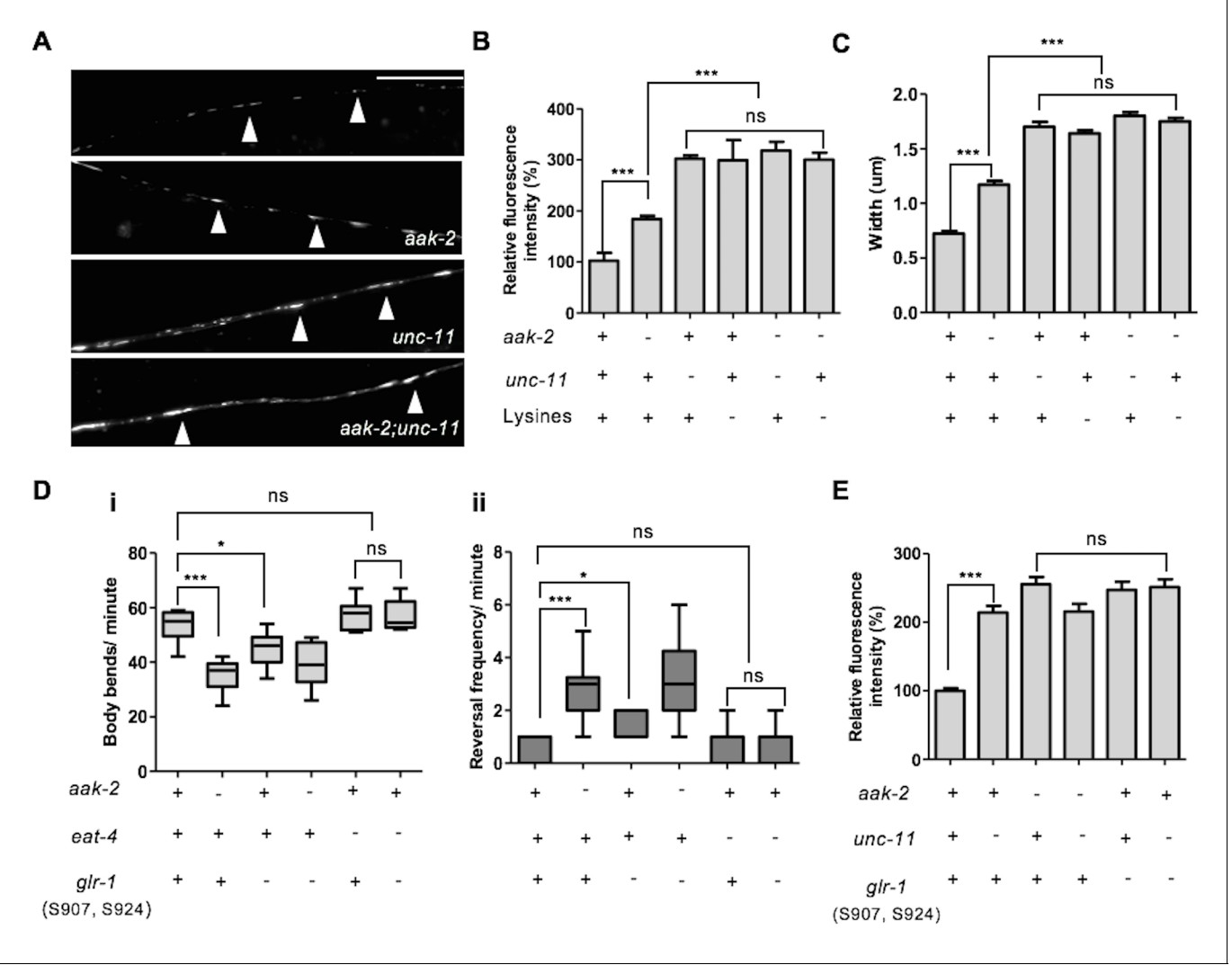

**Figure 6.** AMPK directly regulates GLR-1 abundance through endocytic pathway. (A) Representative images of [P*glr-1::glr-1::gfp*] in the ventral nerve cord in starved WT, *aak-2*, *unc-11* and *aak-2; unc-11* mutants. Scale bars are 10 μm. (B,C) Similar to starved *aak-2* mutants, GLR-1::GFP level (B) and puncta width (C) are increased upon removal of *unc-11* or mutation of four lysines required for GLR-1 ubiquitination and endocytosis [P*glr-1::glr-1(4KR):: gfp*]. Increased GLR-1::GFP level and enhanced puncta width are not further enhanced upon depletion of *aak-2* in GLR-1 endocytosis defective mutants (n>25), Error bars represent ± SEM (one-way ANOVA ***$p<0.0001$). (D) Mutating AMPK phosphorylation sites S907 and S924 to non-phosphorylable alanine residues present in the cytoplasmic domain of GLR-1 (S907A, S924A) results in increased reversal frequency compounded with reduced forward locomotion and this defect can be rescued by introducing *eat-4* mutations (n>10), (one-way ANOVA *$p<0.05$, **$p<0.001$, ***$p<0.0001$). In the box and whisker plots the central line is the median, the edges of the box are the 25th and 75th percentiles, and the whiskers extend to the most extreme data points. (E) Expression of the non-phosphorylable variant of [P*npr-9::glr-1*(S907A, S924A)*::gfp*] results in increased GLR-1::GFP level in the AIB neuronal process and it is not further increased upon disruption of endocytosis in *unc-11* mutants. At least 3 transgenic lines expressing non-phosphorylable variant of GLR-1 (S907A, S924A) were separately examined for this experiment (n>15), Error bars represent ± SEM (one-way ANOVA ***$p<0.0001$).

The following source data is available for figure 6:

**Source data 1.** Analysis of GLR-1 abundance in endocytosis defective mutants.

Elevated glutamatergic transmission within the command interneurons is associated with a hyper-reversal phenotype and a dramatic decrease in the duration of forward movement (*Zheng et al., 1999*). It would therefore be intuitive that an increase in synaptic strength, or input, would result in a consequential increase in reversal frequency. Although we were unable to directly examine the role of AMPK in the regulation of synaptic strength, taken together, our data suggest that AMPK

modulates synaptic strength by regulating GLR-1 abundance in the postsynaptic puncta to bias the behavioural readout in response to starvation.

## AMPK modulates MGL-1 in the mRNA and protein levels in the AIY interneurons to modify distal exploratory behaviour in response to starvation

The results of our epistasis analysis between *mgl-1* and *aak-2* are consistent with *mgl-1* acting downstream of *aak-2* to regulate the transition from local to distal exploration in parallel with *glr-1* in response to starvation. Therefore, to further explore the mechanism by which AMPK regulates MGL-1 activity in AIY, we determined the fluorescence intensity of a [P*mgl-1*::*mgl-1*::GFP] translational fusion protein in both well-fed and starved animals. Starved animals all showed reduced MGL-1 levels, which was completely reversed in *aak-2* mutants (*Figure 7A,B*, *Figure 7—figure supplement 1A*). Moreover, expressing *aak-2* specifically within the AIY interneurons was sufficient to restore MGL-1 levels to those of starved WT animals suggesting that AMPK activity modulates MGL-1 abundance in the AIY in response to starvation.

To determine how AMPK affected the MGL-1 abundance in the AIY we first measured the mRNA levels in well-fed and starved animals using semi quantitative RT-PCR, while in parallel we also monitored *glc-3* levels which are expressed in the AIY where it acts as a glutamate-gated chloride channel required for local search behaviour (*Chalasani et al., 2007*). Although we did not observe any change in the levels of *glc-3* mRNA in well-fed or starved animals, we did observe an slight increase in *mgl-1* mRNA levels in well-fed and a more significant increase in starved *aak-2* mutants (*Figure 7—figure supplement 1B,C*) suggesting that AMPK regulates MGL-1 levels by either blocking its transcription or affecting the stability of *mgl-1* transcripts. The former possibility is less plausible given the short time delay that occurs between the response to food deprivation and the consequent behavioural changes. Since we did not detect any significant change in the *mgl-1* mRNA levels in WT animals, it is likely that AMPK controls the abundance of *mgl-1* transcripts in the AIY and possibly other *mgl-1*-expressing neurons in response to starvation, typical of its function in adapting to energy stress. These results however do not exclude the possible role of AMPK in regulating MGL-1 protein levels directly through phosphorylation, as has been described for some ion channels (*Lang and Föller, 2014*), although this would be in addition to its observed impact on *mgl-1* transcript abundance.

To further investigate this possibility, we mutated the conserved potential AMPK phosphorylation site in MGL-1 (*Figure 7—figure supplement 1D*) to examine if AMPK also regulates MGL-1 through direct phosphorylation. We noted that mutating serine 234 to alanine (S234A) resulted in a modest, but nevertheless significant reduction in forward locomotion (*Figure 7C*), although this mutation did not result in increased MGL-1 levels (*Figure 7—figure supplement 1E*). This suggests that AMPK regulates *mgl-1* activity by impinging on both mRNA and protein function. The AMPK-mediated phosphorylation of serine 234 in MGL-1 may be required for the modulation of distal exploratory behaviour independent of its regulation of MGL-1 levels. We currently cannot exclude that this post-translational modification could affect some aspect of MGL-1 receptor function. Taken together, our results are consistent with a role for AMPK in modulating *mgl-1* function in the AIY interneurons whereby *mgl-1* activity is attenuated in response to starvation-induced AMPK signalling resulting in increased AIY activity and the consequent suppression of reversals.

Our results so far suggest that increased GLR-1 and MGL-1 abundance in the AIB and AIY interneurons, respectively, cause the defective distal exploration in starved *aak-2* animals. To further test whether indeed this may be the case we overexpressed GLR-1 and MGL-1 in the AIB and AIY interneurons, respectively, in WT animals to verify if this misexpression could recapitulate the defective distal exploration of starved *aak-2* mutants. The overexpression of MGL-1 in the AIY resulted in a phenotype that was quite similar to that seen in starved *aak-2* mutants whereas overexpression of GLR-1 in the AIB resulted in a more modest, but nevertheless significant, defect in transition between local to distal exploration. This difference can be explained by the fact that the overexpressed GLR-1 is most likely still targeted for endocytosis and degradation while AMPK is present and active, while in *aak-2* mutants there is a defect in GLR-1 endocytosis allowing it to accumulate to functionally critical levels in the puncta. Nevertheless, the overexpression of either of these transgenes results in a behavioural defect which phenocopies that of starved *aak-2* mutants (*Figure 8—figure supplement 1*).

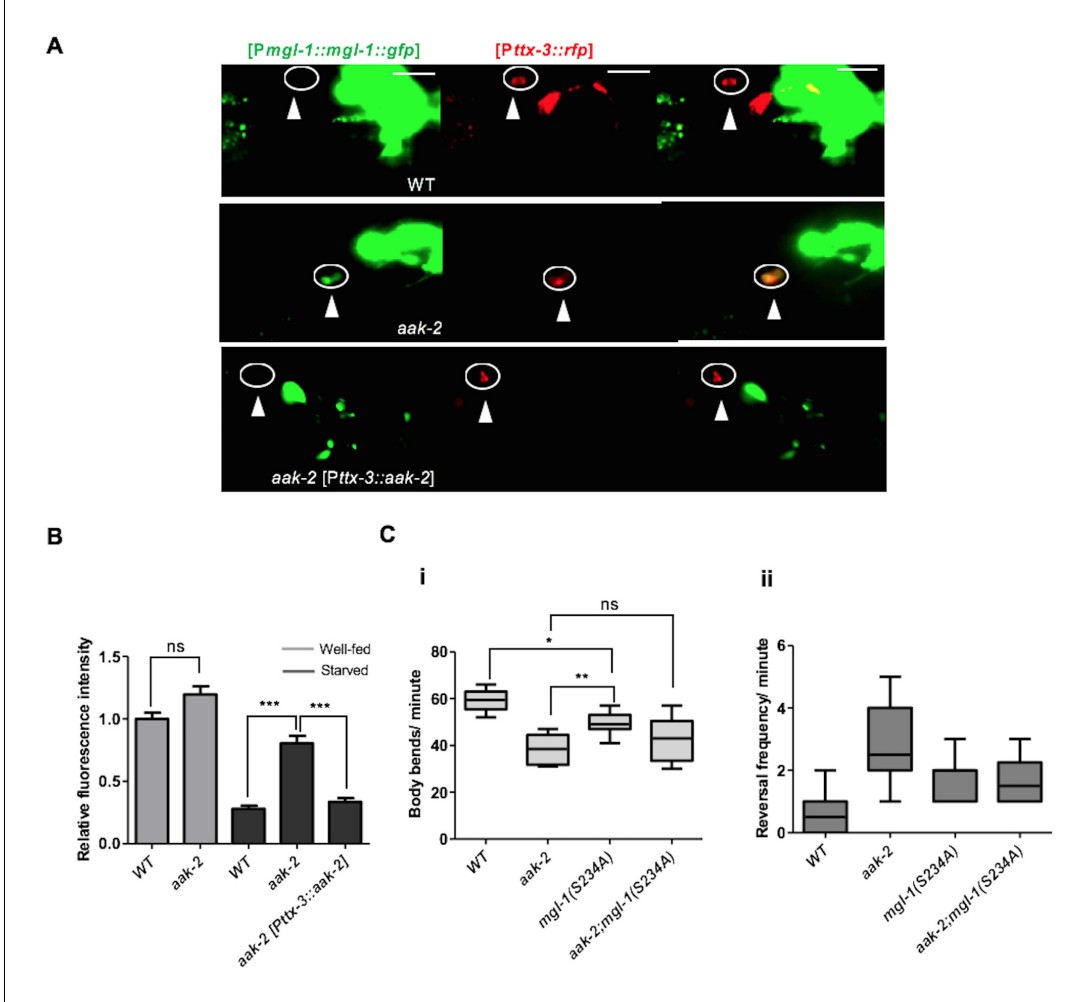

**Figure 7.** AMPK regulates the abundance of MGL-1 levels, while direct phosphorylation may affect MGL-1 function. (A,B) Starvation reduces [MGL-1:: GFP] levels significantly in the AIY [Pttx-3::RFP] of WT animals and to a lesser extent in *aak-2* mutants (AIY neuron is shown in the circle). Increased [MGL-1::GFP] levels in starved *aak-2* mutants can be reversed by targeted expression of *aak-2* in the AIY interneurons using [Pttx-3::aak-2] transgene (n>15), Error bars represent ± SEM (two-way ANOVA ***p<0.0001). Scale bars are 20 μm. (C) Mutating the consensus AMPK phosphorylation site at serine 234 to a non-phosphorylable alanine in MGL-1 (S234A) resulted in a reduced forward locomotion rate that is not further reduced in *aak-2; mgl-1* (S234A) double mutants (i) (n>10), (one-way ANOVA *p<0.05, **p<0.001). In the box and whisker plots the central line is the median, the edges of the box are the 25th and 75th percentiles, and the whiskers extend to the most extreme data points.

The following source data and figure supplement are available for figure 7:

**Source data 1.** Analysis of MGL-1 abundance in WT and aak-2 mutants.

**Figure supplement 1.** AMPK regulates MGL-1 by modulating its steady state mRNA levels while also affecting MGL-1 function through direct phosphorylation.

## Discussion

In most free-living organisms energy stress is often accompanied by characteristic compensatory behaviours that are triggered to increase the probability of encountering a food source (*Wang et al., 2005*; *Gray et al., 2005*; *Dietrich et al., 2015*). The spectrum of these responses however varies dramatically and naturally reflects both the environment and the physiology of the organism. Humans are characteristically hypersensitive and/or rapidly irritable during periods of acute food deprivation providing a familiar example of how a collection of neural circuits become reproducibly activated or blocked in response to this specific acute stress, culminating in highly

predictable behavioural responses. These behavioural changes are achieved in part through synaptic plasticity, which is required for adaptation to varying circumstances in an experience-dependent manner (*Bito, 2010*; *Collingridge et al., 2010*; *Liu et al., 2012*). During our work we have taken advantage of a common physiological response to an acute environmental stress in order to delineate the genetic pathway, the corresponding neuronal circuitry, and the biochemical targets that drive adaptive exploratory behaviours in response to starvation in *C. elegans*.

## AMPK function within the AWC-AIB-AIY circuitry is required for modification of distal exploratory behaviour in response to acute starvation

We and others found that the disruption of AMPK signalling resulted in marked changes in exploratory behaviour partly by reduced forward locomotion (*Lee et al., 2008*; *Cunningham et al., 2014*). However, we also noticed that starved *aak-2* animals execute frequent reversals, which might contribute significantly to their defective distal exploration given the antagonistic relationship between the well-characterized forward and backward locomotory circuits. We were able to improve forward locomotion by the targeted expression of *aak-2* in the AIB and AIY interneurons, while consequently reducing the abnormally frequent reversal behaviour typical of AMPK mutants, consistent with the involvement of AIB and AIY in regulating both reversal frequency and the duration of forward movement (*Tsalik and Hobert, 2003*; *Wakabayashi et al., 2004*). Notably, unlike *aak-2* expression in the sensory neurons, its expression within the AIB and AIY interneurons also improved the chemotaxis defect of *aak-2* mutants suggesting that AMPK may be critical for the integration of sensory signals received by AIB and AIY interneurons.

A recent study showed that *aak-2* compromise mimics the effect of elevated serotonin on movement (*Cunningham et al., 2014*). The AIB and AIY interneurons synapse directly or indirectly onto the command interneurons to control whether *C. elegans* moves forward or backward (*Gray et al., 2005*; *Chalasani et al., 2007*; *Piggott et al., 2011*; *Chen et al., 2013*). Moreover, these neurons have been identified as potential targets of serotonergic signalling to mediate the effects of serotonin on locomotion (*Harris et al., 2009*). It is likely therefore that *aak-2* acts within the AIB and AIY interneurons to modulate the effects of serotonergic signalling on the resulting movement rather than functioning directly within the serotonergic neurons per se.

## AMPK acts as a molecular trigger in the AWC-AIB-AIY circuitry to modify their neuronal activity and the resulting behavioural outputs

The very short time delay that occurs between the response to the lack of food and the observed behaviours makes it unlikely that the effects of AMPK within the AIB and AIY interneurons are mediated by transcription. Alternatively, AMPK has been demonstrated to act directly upon ion channels and gap junctions (*Hallows et al., 2000*; *Lang and Föller, 2014*). Our calcium imaging data are consistent with a model whereby starvation-induced AMPK activation leads to the inhibition of neuronal activity within the AIB, while promoting activity in the AIY interneuron, thereby altering behavioural outputs in a coordinated manner.

How then can we account for the AMPK-dependent starvation-induced changes in calcium levels within the AIB and AIY interneurons and the effects on their respective neural activity? Neuronal activity results from the interplay between synaptic excitation and inhibition. In the brain, excitation is mainly achieved through glutamatergic transmission (*Bito, 2010*; *Collingridge et al., 2010*; *Liu et al., 2012*). Remarkably, our results delineate a novel role for AMPK in the modulation of neuronal activity in response to starvation through the simultaneous regulation of glutamatergic inputs mediated by two different receptors: GluR/GLR-1 in the AIB and MGL-1 in the AIY. Of note, unlike *mgl-1* null mutations that resulted in the partial rescue of locomotory behaviour of *aak-2* mutants, we noticed a complete rescue upon removal of *eat-4* or *glr-1*. As *eat-4* is the only identified vesicular glutamate transporter in *C. elegans*, inhibition of *eat-4* is expected to result in a complete inactivation of both *mgl-1*- and *glr-1*-dependent outputs that are both potentially dependent on glutamate release. In the case of *glr-1*, AIY may require either direct or indirect synaptic inputs from an additional *glr-1*-expressing neuron(s), or alternatively may signal to a *glr-1*-expressing neuron(s) so that eliminating *glr-1* not only results in reduced AIB activity, but also directly or indirectly blocks AIY synaptic inputs or outputs.

## AMPK coordinates glutamatergic signals through distinct mechanisms

In addition to the plethora of processes that are regulated downstream of AMPK activation, AMPK has been shown to modulate synaptic aging (*Samuel et al., 2014*). In mice, starvation increases the activity of the AgRP neurons by affecting their firing rate, ultimately inducing intense food-seeking behaviour and increased feeding (*Dietrich et al., 2015*). The enhanced firing rate of the AgRP neurons has been shown to be largely dependent on glutamatergic inputs mediated by glutamatergic ionotropic AMPA and NMDA receptors (*Bito, 2010*; *Liu et al., 2012*). AMPK modulates this food intake behaviour through the regulation of glutamate release in both mice and in *C. elegans*, although the precise mechanism has not been elucidated in either case (*Yang et al., 2011*; *Cunningham et al., 2012*).

Our data demonstrate that AMPK (*aak-2*) regulates the GLR-1 and MGL-1 abundance postsynaptically. The AWC is sensitive to food availability and is inhibited during starvation, resulting in the inactivation of AIB and the activation of AIY (*Gray et al., 2005*; *Chalasani et al., 2007*). Our data demonstrate that the loss of AMPK signalling results in an increase in AIB neuronal activity and a concomitant decrease in AIY activity. This may be rationalized by the fact that *C. elegans* neurons release neurotransmitter in a graded fashion. Like vertebrate rod and cone photoreceptors which show tonic activity in the dark, and are hyperpolarized or depolarized by light, and removal of light respectively, AWC olfactory neurons have been shown to display basal activity in the absence of odour, but are inhibited or stimulated by odour or odour removal, respectively (*Chalasani et al., 2007*). Therefore, the increased GLR-1 and MGL-1 abundance in the AIB and AIY interneurons could potentially result in responses of longer duration and increased strength upon tonic neurotransmitter release at rest, or upon activation of AWC consequently increasing reversal frequency and shortening runs (*Figure 8*).

In *C. elegans* an increase in glutamatergic inputs mediated by GLR-1 results in an enhanced reversal frequency making this simple behaviour a faithful determinant of increased synaptic strength (*Zheng et al., 1999*; *Burbea et al., 2002*; *Chalasani et al., 2007*). We provide several lines of evidence that demonstrate that AMPK regulates GLR-1 abundance, by affecting its ubiquitylation status during conditions of acute starvation. First, the total GLR-1 level is decreased upon starvation. Second, the amplitude and size/width of GLR-1 puncta that are two determinants of GLR-1 abundance are increased in the AIB neuronal process and in the ventral nerve cord in *aak-2* mutants. Third, this elevated GLR-1 accumulation cannot be further enhanced upon the disruption of the various processes required to remove GLR-1 from the membrane suggesting that AMPK and the ubiquitin-mediated internalization of GLR-1 may therefore function in a linear genetic pathway. Fourth, mutating the AMPK phosphorylation sites in GLR-1 results in increased GLR-1 abundance and defective distal exploratory behaviour. Finally, this defect is comparable to that observed in *aak-2* mutants, while crossing the GLR-1 (S907A, S924A) variant into the *aak-2* mutant background did not further exacerbate the distal exploratory behavioural defect. Therefore, consistent with its well-described role in the mouse where it regulates glutamatergic neurotransmission (*Yang et al., 2011*), AMPK is required to regulate the abundance of GLR-1 in specific glutamatergic neurons. In doing so it modulates glutamatergic inputs and synaptic strength to elicit appropriate behavioural outcomes in response to starvation.

AMPA receptors are either inserted or removed from postsynaptic membranes in an activity-dependent manner leading to the potentiation or depression of synaptic transmission, respectively (*Carroll et al., 1999*). Our findings might be further extended to implicate AMPK in the control of synaptic plasticity that underlies learning and memory as the appropriate regulation of glutamate receptor abundance is critical for long-term memory through habituation in *C. elegans* (*Rose et al., 2003*). Any increase in GluR abundance, and hence signalling, that arises in AMPK-compromised animals is very likely to result in irregular plasticity and abnormal behavioural outcomes, consistent with what is observed in starved AMPK mutants.

In addition to its regulation of GLR-1 abundance in the AIB, we show that AMPK also regulates glutamatergic inputs by regulating MGL-1 in the AIY. A key function of group II mGlu receptors is to modulate neuronal excitability and synaptic transmission at least partly by inhibiting adenylate cyclase (*Dillon et al., 2006*; *Niswender and Conn, 2010*). Group II mGlu receptors are required in both cognitive and emotional processes, and have been linked to various neuropsychiatric conditions, including anxiety, stress-related disorders, schizophrenia and substance misuse

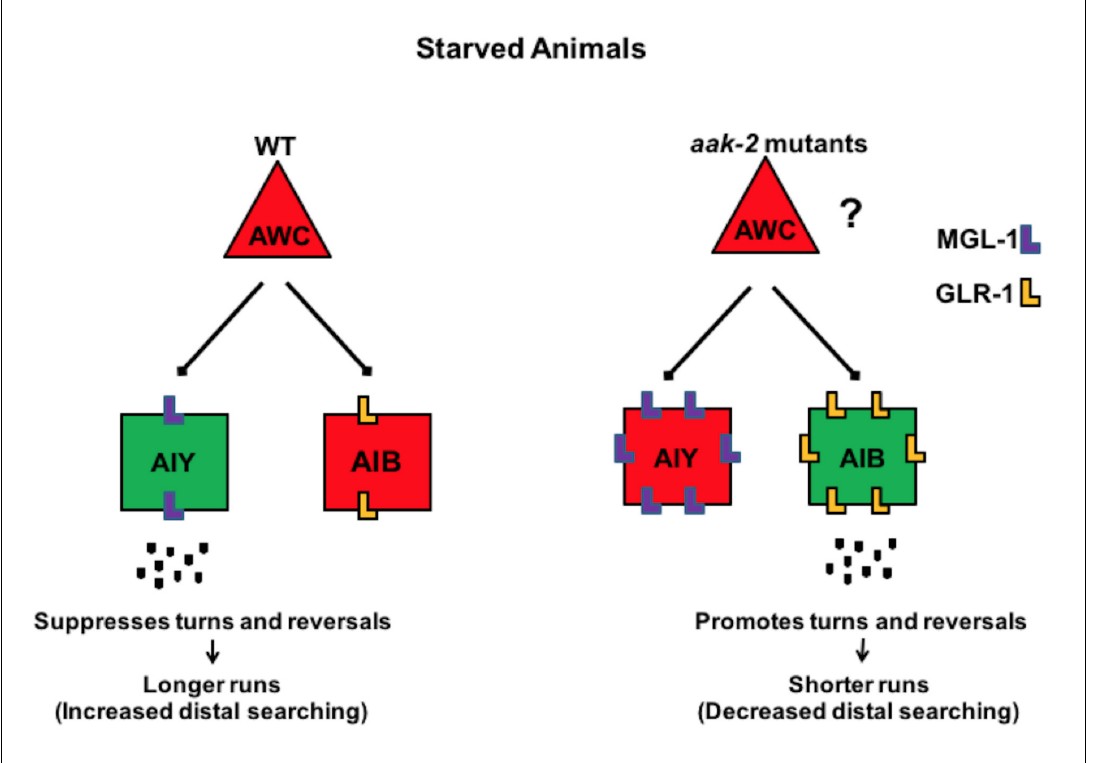

**Figure 8.** AMPK coordinates neuronal activity of the AIB and AIY interneurons by regulating their glutamatergic synaptic inputs in response to acute starvation. The schematic showing the proposed role of AMPK in the AIB and AIY interneurons for regulation of appropriate transition between local to distal exploration during starvation conditions. Green and red colos show depolarized and hyperpolarized neurons, respectively. AMPK regulates GLR-1-mediated synaptic inputs into AIB interneurons by directly phosphorylating GLR-1 and subsequently modulating its endocytosis and potentially its degradation leading to decreased AIB activity in starved animals. AMPK also regulates the MGL-1-dependent synaptic inputs into the AIY interneurons, not only through its effects on *mgl-1* mRNA levels but also by direct phosphorylation of MGL-1 protein. These functions of AMPK within the AIB and AIY interneurons collectively result in the transition between local exploration that occurs in well fed animals to distal exploration that occurs in starved animals, thus allowing them to explore their environment more extensively for energy resources. Legends to figure supplements

The following source data and figure supplement are available for figure 8:

**Source data 1.** glr-1 and mgl-1 over-expression data.

**Figure supplement 1.** Overexpression of *glr-1* and *mgl-1* in the AIB and AIY, respectively results in the similar defect in distal locomotory behaviour as *aak-2* mutants.

(***Niswender and Conn, 2010***). In mammals, activation of mGluR II induces long term depression at GABAergic synapses in the cochlear nucleus magnocellularis neurons where they have been shown to modulate synaptic plasticity (***Lu, 2007***).

In *C. elegans*, *mgl-1* regulates a systemic starvation response and fat accumulation (***Greer et al., 2008***; ***Kang et al., 2009***). Consistent with the inhibitory role of group II mGlu receptors in the vertebrate nervous system, *mgl-1* was recently demonstrated to coordinate environmental sensory cues to modulate the activity of the pharyngeal neural network (***Dillon et al., 2015***). Although some studies have indicated that metabotropic glutamate receptors can be phosphoregulated by kinases such as PKC (***Kim et al., 2005***; ***Ko et al., 2012***), our work has uncovered an AMPK-mediated modulation of the MGL-1 in response to acute food deprivation in *C. elegans*. This effect is at least partly achieved by influencing the regulation of *mgl-1* mRNA levels in starved animals. Considering the brief delay between stimulus and response it is most likely that AMPK mediates its effect on *mgl-1* transcript levels through the phosphorylation of key regulators of *mgl-1* mRNA stability as described previously (***Yun et al., 2005***). Therefore, in addition to its well-described role in the regulation of

ionic channels by affecting their degradation, AMPK may regulate its downstream neuronal effectors by affecting mRNA stability to regulate protein levels.

In summary, our findings suggest a simple model, in which AMPK coordinately regulates the activity of the AIB and AIY interneurons by modulating their glutamatergic synaptic inputs through two distinct mechanisms. AMPK modulates glutamatergic inputs in both interneurons by impinging on AMPA-type glutamate receptor GLR-1 and metabotropic glutamate receptor MGL-1 which consequently regulate distinct downstream behavioural outputs. The most salient feature of our study is our demonstration that AMPK acts as a molecular switch in the nervous system that is most likely not limited to the circuitries engaged in the regulation of feeding and locomotory behaviours. Given the conserved AMPK phosphorylation sites in GLR-1 and MGL-1, we speculate that AMPK regulates the activity of neural circuits in various organisms at least partly by affecting these two receptor types to modulate a wide range of neuronal outputs that respond to physiological or developmental contexts involving various stresses. Alternatively, its starvation-inducible or pharmacological activation could provide a highly effective, non-invasive means of modifying behavioural outputs.

## Materials and methods

### Strains

*Caenorhabditis elegans* strains were maintained under standard conditions (*Brenner, 1974*). The Bristol isolate (N2) was used as wild type. The following alleles were used in this study: *aak-1 (tm1944), aak-2(ok524), eat-4(ky5), glr-1 (n2461), lite-1(ce314), unc-11(e48), mgl-1(tm1811), osm-6 (p811), avr-14(ad1305), nmr-1(ak4), mgl-2(tm355), sid-1(pk3321)*. Please see **Supplementary file 1** for the complete list of strains used in this study.

### Plasmid constructs and transgenes

Genomic *aak-2* and *mgl-1* and their 2 kb and 3 kb upstream sequences were amplified and cloned upstream of GFP to generate [P*aak-2::aak-2::gfp*] and [P*mgl-1::mgl-1::gfp*]. *unc-119, unc-54, glr-1, tph-1, rig-3, ttx-3, npr-9, che-2* promoters were amplified from genomic DNA and were cloned upstream of *aak-2* cDNA (*Narbonne and Roy, 2009*) to drive *aak-2* expression globally throughout the nervous system, body wall muscle and in different neuronal subtypes. GLR-1::GFP was amplified from *nuIs24* strain and was inserted downstream of *npr-9* promoter to generate [P*npr-9::glr-1::gfp*]. The resulting constructs were injected at 10–50 ng/µl and either *Punc-122::gfp, Pelt-2::gfp, Pmyo-2::gfp* or *Punc-122::dsred* were used as the co-injection marker. To generate the GLR-1 variants including (S907A) and (S924A), PCR-mediated site-directed mutagenesis was performed using Gene-Tailor site-directed mutagenesis (Invitrogen, Carlsbad, CA) on [P*npr-9::glr-1::gfp*] and transgenic lines were generated after microinjecting plasmids at 50 ng/µl.

We used CRISPR/Cas9 to mutagenize the AMPK phosphorylation site serine 234 to alanine in *mgl-1* as described previously (*Paix et al., 2014*).

### Quantitative analysis of exploratory behaviour

Well-fed adult animals were scored for exploratory behaviour following 2 hr starvation as described previously (*Gray et al., 2005*; *Calhoun et al., 2015*). For our time course experiment mid-L4 animals were scored for exploratory behaviour either immediately after removal from food or following longer durations off food. To minimize the behavioural modifications caused by variability in their environment, animals were grown on the similar size and amount of bacterial patches (*Calhoun et al., 2015*). All body bends and reversals were scored by eye, by an investigator blind to the genotype of the animals.

L4 animals of *sid-1(pk3321); uIs69 [pCFJ90 (pmyo-2*::mCherry) + *unc-119p::sid-1*] and *aak-2 (ok524); sid-1(pk3321); uIs69 [pCFJ90 (pmyo-2*::mCherry) + *unc-119p::sid-1*] which are nervous system RNAi hypersensitive were transferred to the RNAi plates seeded with overnight cultures of HT-115 *E. coli* clones expressing double stranded RNAi targeting *glr-1, glr-2, eat-4, mod-1, ser-2, inx-7, unc-7, inx-19, gar-2, gcy-1, flp-1, mgl-1, npr-11*. The exploratory behaviour of the resultant progeny was monitored upon reaching to adulthood.

### Locomotory behaviour in the presence of food

The basal and enhanced slowing response was measured by transferring well-fed animals to plates with no bacteria, and transferring them to an assay plate with a thin layer of bacteria immediately or after 30 min off food as described previously (*Sawin et al., 2000*).

### Chemotactic analysis

Chemotaxis assays towards a bacterial lawn or isoamyl alcohol (IAA) were performed on 10 cm nematode growth medium (NGM) plates incubated at room temperature (22°C) overnight as previously described. Animal behaviour was scored by eye, by an investigator blind to the genotype of the animal. 10 and 300 animals were used for each chemotaxis assay, either toward the bacterial lawn and IAA, respectively (*Hart, 2006*).

### Optogenetics

C. *C.elegans* grown on NGM plates supplied with 5 µM all-trans retinal (ATR) were starved for 2 hr and their exploratory behaviour was monitored on retinal-free NGM plates. A 1 min pulse of blue (480 nm) or yellow light (540 nm) was delivered from an Arc lamp (EXFO) by a 10× objective (Zeiss M2Bio) to the head of an individual animal to activate ChR2 or Arch, respectively as described previously (*Piggott et al., 2011*; *Kocabas et al., 2012*). In order to eliminate intrinsic phototaxis responses, we used *lite-1* mutants that are defective in UV light phototaxis. *lite-1* mutants did not show any obvious defect in the locomotory behaviour in our assays. To depolarize the AIY interneurons or hyperpolarize the AIB interneurons, we used previously described strains expressing channelrhodopsin-2 (ChR2) or archaerhodopsin-3 (Arch), respectively (*Kocabas et al., 2012*).

### In vivo imaging

For the calcium imaging adult transgenic animals expressing [P*npr-9*::G-CaMP3+ P*npr-9*::DsRed2B] (*Piggott et al., 2011*) or [P*ttx-3*::G-CaMP1+ unc-122::GFP] (*Chalasani et al., 2007*) were transferred to a 10% agarose pad following 2 hr starvation. A coverslip was then applied and the calcium transients were monitored in the cell body or neuronal process of the AIB and AIY interneurons, respectively as previously described (*Lemieux et al., 2015*). Briefly, to measure calcium transients in the AIB interneurons fluorescence images (green channel: G-CaMP3, red channel: DsRed2B) were acquired sequentially over 120 s window at a frequency of 2 Hz through a 63X objective on a Zeiss microscope. To measure calcium transients in the AIY interneurons fluorescence images (Green channel) were acquired from the neuronal process in the nerve ring over a 65 s window at a frequency of 3 Hz through a 63X objective. The resulting time-point series obtained from the fluorescence images for the AIB and AIY were then analyzed using ImageJ (RRID:SCR_001935).

To calculate the baseline values ($F^b$) averages of global minima for 5–10 frames over the duration of each time-lapse sequence were obtained and the difference between a given each time point's fluorescence and $F^b$ ($\Delta F$) was normalized by dividing $\Delta F$ to $F^b$ to plot $\Delta F/F^b$ for each individual. In our analyses, we report the total integrated signal of all the observed peaks during the 65 s or 120 s recording window that consists of an unambiguously defined initiation of a spontaneous transient in the neural process or cell body for the AIY and AIB, respectively. To measure the total integrated fluorescence the $\Delta F/F$ for each time point were summed over each time-lapse sequence to obtain a measurement for each time series sequence as described previously (*Lemieux et al., 2015*).

Calcium imaging in freely behaving animals was performed on a Zeiss microscope equipped with a Hamamatsu camera as previously described (*Luo et al., 2014*; *Flavell et al., 2013*). To prevent behavioural responses to blue light, the imaging was performed in the *lite-1* background (*Flavell et al., 2013*). Animals were placed on 10% agarose pads to slow down their locomotion and the pad was sealed in a small chamber to prevent evaporation. The elapsed time from placing animals on the agarose pads to initiation of imaging was <5 min. Fluorescence time-lapse imaging was performed over 90 s at a frequency of 2 Hz using a 10x objective by an experimenter blind to the genotype. The images were processed as described before (*Lemieux et al., 2015*; *Luo et al., 2014*).

### Microscopy

All images were acquired using a Zeiss AX10 microscope equipped with a Hamamatsu camera as previously described (*Burbea et al., 2002*). Starved GLR-1::GFP-expressing strains were immobilized

with 10 mM levamisole. Images were captured and processed using AxioVision software. Maximum intensity projections of Z series stacks were obtained. GLR-1 levels were calculated using imageJ and normalized relative to WT. Puncta widths were estimated as the peak width at half-maximal amplitudes as described previously (*Burbea et al., 2002*). At least three transgenic lines were examined for these experiments and all the imaging and subsequent quantification was performed by an experimenter who was blind to the genotype.

The extent of signal overlaps between GLR-1 and the presynaptic marker EAT-4 was indicated by imaging transgenic animals expressing GLR-1::GFP and EAT-4::mCherry. Maximum intensity projections of sequential images from fluorescent green and red filters were obtained. The local maximum in each punctum was used to determine the spatial position of the GFP and mCherry peaks. The percentage of GLR-1 puncta co-localizing with a EAT-4 punctum was assessed by the fraction of GLR-1 puncta peaks that were less than 1 μm (or the width of an average punctum) from a presynaptic peak as previously described (*Burbea et al., 2002*).

For quantification of MGL-1 abundance in the AIY, the fluorescence intensity was measured in at least 3 transgenic lines expressing [*Pmgl-1::mgl-1::gfp*] which were starved for two hours. The exposure time was adjusted in the well-fed wild-type background and images were acquired by an experimenter who was blind to the genotype. Pixel intensity was measured in ImageJ (NIH) by calculating the mean pixel intensity for the entire region of interest.

## Western blot

To compare GLR-1 levels, proteins were extracted from well-fed and starved WT and *aak-2* mutants by sonication in lysis buffer (50 mM Hepes pH7.5, 150 mM NaCl, 10% glycerol, 1% Triton X-100, 1.5 mM MgCl2, 1 mM EDTA and protease inhibitors). Protein concentration was then determined by NanoDrop 2000c spectrophotometer (Thermo Scientific, Waltham, MA). A similar amount of protein for each condition was then subjected to 8% SDS-PAGE and then transferred to a nitrocellulose membrane (Bio-Rad, Canada) and blotted by anti-GFP antibody (RRID:AB_162553) as described previously (*Xie and Roy, 2015*).

## Semi-qRT-PCR

RNA extraction was performed in well-fed and starved animals as described previously (Lu and Roy. Similar quantities of RNA from each sample were used to generate comparable amplicon levels for each gene tested. PCR was performed using gene-specific primers for each query gene for 15 cycles with ProtoScript M-MuLV *Taq* RT-PCR Kit (NEB, E6400S).

## Acknowledgements

We would like to thank Cori Bargmann, Sreecanth Chalasani and Joshua Kaplan for reagents, along with Michael Hendricks for both reagents and very helpful discussion. We are grateful to our fellow lab members, and I am particularly indebted to my colleagues Francois Fagotto and Monique Zetka for their support. We also thank Mohammad R Raeesi N for his help with image processing and analysis and also the Caenorhabditis Genetic Center for *C. elegans* strains. This work was supported by funds awarded by the Canadian Institutes of Health Research (CIHR).

## Additional information

### Funding

| Funder | Grant reference number | Author |
| --- | --- | --- |
| Canadian Institutes of Health Research | MOP 84486 | Richard Roy |

The funders had no role in study design, data collection and interpretation, or the decision to submit the work for publication.

## Author contributions
MA, Conception and design, Acquisition of data, Analysis and interpretation of data, Drafting or revising the article; RR, Conception and design, Analysis and interpretation of data, Drafting or revising the article

## Author ORCIDs
Richard Roy, http://orcid.org/0000-0003-3001-9311

## Additional files

### Supplementary files
• Supplemenmtary file 1. Complete list of strains used in this study.

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
