## [Decision Letter]

Thank you for submitting your article "AMPK acts as a molecular trigger to coordinate glutamatergic signals and adaptive behaviours during acute starvation" for consideration by *eLife*. Your article has been reviewed by two peer reviewers, and the evaluation has been overseen by a Reviewing Editor and K VijayRaghavan as the Senior Editor. The reviewers have opted to remain anonymous.

The reviewers have discussed the reviews with one another and the Reviewing Editor has drafted this decision to help you prepare a revised submission.

In this paper, Ahmadi and Roy address how the AMP-activated protein kinase (AMPK, *aak-2*) regulates behavioral response to acute starvation. Because AMPK monitors the status of cellular energy, the findings provide a mechanistic link between physiological and behavioral responses to stress of food-deprivation. The study has identified the key interneurons in a locomotory circuit as the functional targets of the AMPK *aak-2* in response to starvation. It has also found that *aak-2* regulates the stability of two glutamatergic receptors in the interneurons to generate starvation-induced behavioral changes. The study employs multiple experimental approaches to thoroughly support the conclusions.

The following concerns should be addressed:

1) Figure 1 and Figure 1—figure supplement 1. While it seems clear that the *aak-2* mutant displays a defect in suppressing reversals when starved, the defects of *aak-2* in locomotory speed do not seem to be specific to starvation. In Figure 1—figure supplement 1 well-fed *aak-2* also moves more slowly than well-fed wild-type animals. Because *aak-2* is also possibly expressed in motor neurons and some pre-motor neurons, *aak-2* may regulate speed under the normal condition. The effect of *aak-2* on starvation-induced decrease in reversal rate is a parameter that is more specific to starvation, which should be the focus for behavioral characterization.

Along the same line, in Figure 1, the authors showed that starving the animals for 10 mins is already causing a behavioral phenotype. Are *aak-2* animals with no starvation different from wild type? In other words, are the *aak-2* mutant phenotype caused by developmental deficits?

Can overexpression of GLR-1 in AIB or MGL-1 in AIY mimic the effect of *aak-2*?

2) It seems that the calcium imaging results were collected from animals that were immobilized without any sensory stimulation. AIB activity is known to link with reversals, therefore, it is reasonable to assume that the observed and quantified spontaneous activity in AIB represent "attempts" of reversals. However, it is not clear what the spontaneous activity measured in AIY represents. Some clarification is needed.

Can one measure the activity of these neurons during animal movements?

Along the same line, one question is how the imaging trials were selected since it is just spontaneous activity being measured, was the experimenter blinded to the genotype of the animal?

3) There are some concerns about the reagents and the way the receptor imaging experiments are done. Why are the Figure 6 experiments not done in AIB?

Are the strains used to quantify GLR-1 abundance in AIB or MGL1 in AIY stable integrated transgenes? One problem with this type of imaging experiments is the overexpression effect of transgenes.

---

## [Author Response]

*The following concerns should be addressed:*

*1) Figure 1 and Figure 1—figure supplement 1. While it seems clear that the aak-2 mutant displays a defect in suppressing reversals when starved, the defects of aak-2 in locomotory speed do not seem to be specific to starvation. In Figure 1—figure supplement 1 well-fed aak-2 also moves more slowly than well-fed wild-type animals. Because aak-2 is also possibly expressed in motor neurons and some pre-motor neurons, aak-2 may regulate speed under the normal condition. The effect of aak-2 on starvation-induced decrease in reversal rate is a parameter that is more specific to starvation, which should be the focus for behavioral characterization.*

Based on our analysis *aak-2* mutants do not show any difference in their rate of forward or backward locomotion when assayed in the presence of food; a situation where AMPK is most likely inactive (Figure 1—figure supplement 1). Similarly, *aak-2* animals that were previously well-fed or starved display a comparable rate of forward locomotion similarly-treated wild type animals when evaluated on an abundant food source after a short period of time (Figure 1—figure supplement 1). To further examine if *aak-2* mutants display a general dampening of locomotory speed we examined the forward locomotion rate (body bends) of *aak-2* mutants upon exposure to a gentle touch applied to the posterior (just before the anus). We found that *aak-2* mutants show similar locomotory speed compared to WT animals following this touch stimulus suggesting that they do not display any defect in locomotory speed (Figure 1—figure supplement 1 and subsection “Neuronal AMPK signalling triggers distal exploratory behaviour in starved animals”, fourth paragraph). Therefore, it is likely that the basal AMPK activity regulates the observed change in speed specifically in response to food depletion in well-fed animals, while it has a more pronounced effect on behaviour when activated upon prolonged starvation. Since the increased reversal frequency affects the duration of forward movement (Burbea M, Dreier L., 2002; Tsalik and Hobert, 2003 and our Figure 4—figure supplement 2), we believe that the forward locomotion rate in starved *aak-2* animals is very likely to be also affected by the *aak-2*-associated defect in suppressing reversals. This can potentially explain the reason for greater difference in forward locomotion between *aak-2* and WT animals upon 2 hours of starvation compared to 10 minutes off food. Because of the antagonistic relationship between forward and backward locomotion, we always try to include the results for both forward and backward locomotion throughout our study. We pointed this out in the third paragraph of the aforementioned subsection in our description of the data presented in the first figure (Figure 1). We show that the effect of AMPK on locomotory behaviour is largely confined to starved animals, while AMPK also regulates reversal frequency.

*Along the same line, in Figure 1, the authors showed that starving the animals for 10 mins is already causing a behavioral phenotype. Are aak-2 animals with no starvation different from wild type? In other words, are the aak-2 mutant phenotype caused by developmental deficits?*

We examined locomotory behaviour in the presence of food and in response to a mechanical stimulus in the posterior region and we conclude that AMPK mutants are not impaired in any apparent way. *aak-2* animals do display a reduced forward rate of locomotion in the absence of food, but not in the presence of a thin layer of bacteria (Figure 1—figure supplement 1). As previously mentioned, to examine if *aak-2* mutants display a decrease in locomotory speed we monitored the locomotory behaviour of *aak-2* mutants upon exposure to a gentle touch applied to the posterior end of their body. In our opinion this serves as a good control since it is a stimulus that is unrelated to the starvation-sensing circuit. We found that *aak-2* mutants show no difference in locomotory speed compared to WT animals under these conditions (Figure 1—figure supplement 1 and subsection “Neuronal AMPK signalling triggers distal exploratory behaviour in starved animals”, fourth paragraph), thus suggesting that the locomotory phenotypes we describe in our study are unlikely to be a consequence of any more general developmental defect.

In addition, when *aak-2*(RNAi) was performed in late L2 stage larvae, well after the majority of motor neurons are generated, we noted a modest but significant reduction in forward locomotion rate upon removal from food (data not shown) that was compounded with an increase in reversal frequency when the animals were starved (Figure 1—figure supplement 2 and in the aforementioned paragraph) suggesting that the reduced locomotory speed caused by AMPK loss of function in starved animals is unlikely to be a consequence of a more general developmental defect that impinges on motor function. This is further corroborated by our optogenetic results where we show that although AIY activation or AIB inactivation in starved WT animals (when AIB is likely inactive and AIY is likely active (Gray et al., 2005)), do not change their locomotory behaviour, while optogenetically activating AIY or inactivating AIB in *aak-2* mutants ameliorates both forward and backward locomotion. Therefore, taken altogether it is unlikely that the reduced locomotory speed in *aak-2* mutants is a consequence of a more general developmental deficit. Rather, it is most probably a consequence of reduced AIY activity and increased AIB activity in starved *aak-2* mutants.

*Can overexpression of GLR-1 in AIB or MGL-1 in AIY mimic the effect of aak-2?*

This is a great suggestion that could further support our conclusions. We did the experiment and the data have been added to the revised manuscript as Figure 8—figure supplement 1. Previous studies have shown that overexpression of GLR-1 in the AIB interneurons causes an increase in reversal rate in the absence of food(Chalasani et al., 2007), while a group in Japan has recently reported that *mgl-1* mutants display a reduced reversal rate. Their data indicate that the expression levels of MGL-1 in the AIY interneurons are critical for appropriate chemotaxis toward DA and increased MGL-1 levels result in abnormal choice between the repellant and attractant (Suehiro and Mitani, International *C. elegans* Conference 2015). These results are consistent with our data that demonstrate the accumulation of GLR-1 and MGL-1 in the AIB and AIY interneurons of starved AMPK mutants, respectively. Furthermore, these changes in abundance correlate with changes in the forward and backward locomotion rate and aberrant chemotaxis. In our study, using various approaches, we highlight that the appropriate suppression of reversals that would normally occur during starvation is impaired in *aak-2* mutants and is potentially caused by the increased abundance of GLR-1 and/or MGL-1. To determine whether the increased abundance of these receptors can cause the same phenotype as *aak-2* mutants, we examined the locomotory behaviour of starved animals that overexpress GLR-1 in the AIB, or MGL-1 in the AIY, or in animals that overexpress both transgenes. As we have shown in Figure 8—figure supplement 1, overexpressing MGL-1 in the AIY resulted in a phenotype that was quite similar to that seen in starved *aak-2* mutants. The overexpression of MGL-1 in the AIY interneurons was more effective in phenocopying the locomotory defects of starved *aak-2* mutants than the effect of overexpression of GLR-1 in the AIB, which resulted in a modest, but nevertheless significant, defect in transition between local to distal exploration. This can be explained by the fact that overexpressed GLR-1 is most likely still targeted for endocytosis and degradation while AMPK is present and active. In *aak-2* mutants there is a defect in GLR-1 endocytosis allowing it to accumulate in puncta. However, when it is overexpressed there is some accumulation but AMPK is still present to modulate the excess accumulation of the receptor. Nevertheless, overexpression of either of these, or both transgenes results in a similar behavioural defect as that which we report for *aak-2* mutants. We have added these data to the manuscript as Figure 8—figure supplement 1, and we describe our interpretations in the last paragraph of the subsection “AMPK modulates MGL-1 in the mRNA and protein levels in the AIY interneurons to modify distal exploratory behaviour in response to starvation”.

*2) It seems that the calcium imaging results were collected from animals that were immobilized without any sensory stimulation. AIB activity is known to link with reversals, therefore, it is reasonable to assume that the observed and quantified spontaneous activity in AIB represent "attempts" of reversals. However, it is not clear what the spontaneous activity measured in AIY represents. Some clarification is needed.*

Two previous studies which measured the calcium levels in the AIY in freely moving animals had shown that the calcium peaks observed in the AIY correlate with termination of reversals and onset and duration of forward runs with gradual increases preceding forward run initiation. This also coincides with point where the locomotory speed is maximal (Flavell et al., 2013; Luo et al., 2014). This information is added to the new text file (subsection “Parallel and opposing function of AMPK in the modulation of AIB and AIY outputs”, first paragraph).

To further investigate if the defective distal exploration of *aak-2* mutants is a consequence of decreased AIY activity and increased AIB activity in starved *aak-2* mutants and to correlate the locomotory activity with changes in the AIB and AIY neuronal activity, we performed calcium imaging in the AIY and AIB in freely moving WT and *aak-2* animals during starvation. Consistent with previous studies we noted a correlation between the increased Ca^2+^ influxes in the AIY with termination of reversals which remains high during forward runs. Moreover, not surprisingly, the frequency and intensity of Ca^2+^ influxes/neuronal activity in the AIB increases during reversals. These data further corroborate our findings presented in the original manuscript that showed that the defective transition from local to distal exploration during starvation is a consequence of both decreased AIY activity and increased AIB activity. These new data have been added to this revised manuscript as Figure 4—figure supplement 2, while we describe this correlation in the text (subsection “Parallel and opposing function of AMPK in the modulation of AIB and AIY outputs”, first and third paragraphs).

*Can one measure the activity of these neurons during animal movements?*

To examine the correlation between neural activity in AIB and AIY and the locomotory behaviours we describe, we performed calcium imaging in freely moving WT and *aak-2* animals placed on 10% agarose pads as a means to reduce their movement. These data have been included in the revised manuscript as Figure 4—figure supplement 2 this correlation is described in the text (subsection “Parallel and opposing function of AMPK in the modulation of AIB and AIY outputs”, first and third paragraphs).

*Along the same line, one question is how the imaging trials were selected since it is just spontaneous activity being measured, was the experimenter blinded to the genotype of the animal?*

Most of our calcium imaging studies were performed in immobilized animals and the experimenter was indeed blind to the genotype of animals. Moreover, every time that this experiment was performed, all the animals placed on the agarose pad were examined unless the animals were unhealthy or moved out of field of view.

*3) There are some concerns about the reagents and the way the receptor imaging experiments are done. Why are the Figure 6 experiments not done in AIB?*

In Figure 6 we investigated if AMPK might regulate GLR-1 abundance in the nervous system through effects on the endocytic pathway using two different approaches: first, by using *unc-11* mutants which disrupt the AP2-adaptin complex and endocytosis or secondly, using a GLR-1 transgenic variant in which the critical lysine residues that are targeted by ubiquitylation and which trigger endocytosis are mutated to alanine. Since we showed that AMPK plays an important role in the modulation of GLR-1 abundance in the ventral nerve cord by modulating its endocytosis, we wanted to determine if AMPK might regulate GLR-1 abundance specifically in the AIB neuronal puncta in starved animals through similar mechanisms. To test this we used a non-phosphorylable GLR-1 variant that was expressed specifically in the AIB and found that AMPK indeed regulates GLR-1 abundance in the neuronal process through its effects on the endocytic pathway. Although we discussed the results of our analysis of *aak-2; unc-11*, which we initially performed as a control but did not include it in the final figure, we have now incorporated this result into the final figure and we specifically clarify that this experiment was performed in the AIB, and not the ventral nerve cord, both in the text and in the legend for Figure 6.

*Are the strains used to quantify GLR-1 abundance in AIB or MGL1 in AIY stable integrated transgenes? One problem with this type of imaging experiments is the overexpression effect of transgenes.*

In these experiments, all the imaging and subsequent quantifications were performed by an experimenter blind to the genotypes.

In the case of MGL-1 we introduced GFP into the C terminus of *mgl-1* using CRISPR technology. Therefore the GFP signal was initially quantified in the strains that express a single copy of the MGL-1::GFP from endogenous *mgl-1* promoter sequence. However, since *mgl-1* is expressed in the AIY interneurons at much lower levels compared to other *mgl-1*-expressing neurons (data not shown), this signal was very faint and was easily bleached, we repeated and confirmed these results in strains that express MGL-1::GFP in extra chromosomal arrays in order to obtain better quality images and eliminate the problem of rapid photobleaching. To ensure that our observations of the multi-copy transgene are not a consequence of differential expression of the array, 3 independent MGL-1::GFP-expressing strains were examined by an experimenter who was blind to the genotypes of the strains in each case. The results obtained with all three transgenic lines were consistent with our observations made with the CRISPR-introduced allele. We therefore strongly believe that the signal we observe with the extrachromosomal transgenes recapitulates the endogenous situation and indicates that MGL-1 levels vary between starved WT and *aak-2* mutant animals.

Since we had to assess the GLR-1::GFP signal in the neuronal process of AIB, a single copy transgene was not sufficient to generate a strong and quantifiable signal for our experiments. Therefore, as was the case with the MGL-1::GFP, we generated multiple transgenic lines which were scored by an experimenter who was blind to the genotypes while the images were being acquired and also during subsequent analysis. We obtained reproducible results from all of our transgenic lines that were consistent with GLR-1 being regulated by AMPK in the AIB neuronal process. Moreover, since GLR-1 abundance in the AIB neuronal process was clearly elevated in *unc-11* animals, and which was not further enhanced upon the removal of *aak-2*, we conclude that AMPK must be part of a linear pathway that contributes to the endocytic regulation of GLR-1 abundance. Consistent with this possibility GLR-1 variants that lacked the consensus AMPK phosphorylation sites in GLR-1 accumulated in a manner similar to *unc-11* mutants or in *aak-2; unc-11* doubles. Because we can quantify differences in the levels of GLR-1 and its variants in the various genotypes we have tested, it seems unlikely that the transgenic expression levels contribute significantly to the regulation that we observe. We therefore contend that the AMPK-dependent differences in GLR-1 abundance that we observe in the AIB neuronal process in starved animals are not the result of the increased GLR-1 contributed by the extrachromosomal, multicopy transgenes, but rather due to the regulation of the protein by AMPK and additional regulators of the endocytic degradation pathway.